# PAVI:
# PLATE-AMORTIZED VARIATIONAL INFERENCE

## ABSTRACT

Given observed data and a probabilistic generative model, Bayesian inference aims at obtaining the distribution of a model's latent parameters that could have yielded the data. This task is challenging for large population studies where thousands of measurements are performed over a cohort of hundreds of subjects, resulting in a massive latent parameter space. This large cardinality renders off-the-shelf Variational Inference (VI) computationally impractical. In this work, we design structured VI families that can efficiently tackle large population studies. Our main idea is to share the parameterization and learning across the different i.i.d. variables in a generative model -symbolized by the model's *plates*. We name this concept *plate amortization*, and illustrate the powerful synergies it entitles, resulting in expressive, parsimoniously parameterized and orders of magnitude faster to train large scale hierarchical variational distributions. We illustrate the practical utility of PAVI through a challenging Neuroimaging example featuring a million latent parameters, demonstrating a significant step towards scalable and expressive Variational Inference.

## 1 INTRODUCTION

Population studies analyse measurements over large cohorts of human subjects. These studies are ubiquitous in health care (Fayaz et al., 2016; Towsley et al., 2011), and can typically involve hundreds of subjects and measurements per subject. For instance in Neuroimaging (Kong et al., 2019), measurements $X$ can correspond to signals in hundreds of locations in the brain for a thousand subjects. Given this observed data $X$, and a generative model that can produce data given model parameters $\Theta$, we want to recover the parameters $\Theta$ that could have yielded the observed $X$. In our Neuroimaging example, $\Theta$ can be local labels for each brain location and subject, together with global parameters common to all subjects –such as the connectivity corresponding to each label. We want to recover the *distribution* of the $\Theta$ that could have produced $X$. Following the Bayesian formalism (Gelman et al., 2004), we cast both $\Theta$ and $X$ as Random Variables (RVs) and our goal is to recover the *posterior* distribution $p(\Theta|X)$. Due to the nested structure of our applications we focus on the case where $p$ corresponds to a Hierarchical Bayesian Model (HBM) (Gelman et al., 2004). In the context of population studies, the multitude of subjects and measurements per subject implies a large dimensionality for both $\Theta$ and $X$. This large dimensionality in turn creates computational hurdles that we tackle through our method.

Several inference methods exist in the literature. Earliest works resorted to Markov Chain Monte Carlo (Koller & Friedman, 2009), which tend to be slow in high dimensional settings (Blei et al., 2017). Recent approaches, coined Variational Inference (VI), cast the inference as an optimization problem (Blei et al., 2017; Zhang et al., 2019; Ruiz & Titsias, 2019). Inference reduces to finding the distribution $q(\Theta; \phi) \in \mathcal{Q}$ closest to the unknown posterior $p(\Theta|X)$ in a variational family $\mathcal{Q}$ chosen by the experimenter. Historically, VI required to manually derive and optimize over $\mathcal{Q}$, which remains an effective method where applicable (Dao et al., 2021). In contrast, we follow the idea of automatic VI: deriving an efficient family $\mathcal{Q}$ directly from the HBM $p$ (Kucukelbir et al., 2016; Ambrogioni et al., 2021a;b). Our method also relies on the pervasive idea of amortization in VI: deriving posteriors usable across multiple data points, which can be linked to meta-learning (Zhang et al., 2019; Ravi & Beatson, 2019; Iakovleva et al., 2020; Yao et al., 2019). In particular, Neural Processes share with our method the conditioning a density estimator by the output of a permutation-invariant encoder (Garnelo et al., 2018; Dubois et al., 2020; Zaheer et al., 2018). Meta-learning

studies problems with few hierarchies, similar to many VI methods (Ravi & Beatson, 2019; Tran et al., 2017). Agrawal & Domke (2021) notably studied the case of 2-level HBMs, providing with theoretical guarantees. In contrast, our focus is rather computational, and we study generic HBMs with an arbitrary number of hierarchies, aiming at tackling large population studies efficiently.

Modern VI is effective in low-dimensional settings, but does not scale up to large population studies –that involve millions of random variables. In this work we identify and tackle 2 challenges to enable this scale-up. A first challenge with scalability is a detrimental trade-off between expressivity and high-dimensionality (Rouillard & Wassermann, 2022). To reduce the inference gap, VI requires the variational family $\mathcal{Q}$ to contain distributions closely approximating $p(\Theta|X)$ (Blei et al., 2017). Yet the form of $p(\Theta|X)$ is usually unknown to the experimenter. Instead of a lengthy search for a valid family, one can resort to universal density approximators: normalizing flows (Papamakarios et al., 2019). But the cost for this generality is a heavy parameterization, and normalizing flows scale poorly with the dimensionality of $\Theta$. In large populations studies, as this dimensionality grows, the parameterization of normalizing flows becomes prohibitively large. To tackle this challenge, Rouillard & Wassermann (2022) recently proposed –in the ADAVI architecture– to partially share the parameterization of normalizing flows across the hierarchies of a generative model. While ADAVI tackled the over-parameterization of VI in population studies, it still could not perform inference in very large data regimes. This is due to a second challenge with scalability: as the size of $\Theta$ increases, the evaluation of a single gradient over the entirety of an architecture's weights quickly requires too much memory and compute. This second challenge can be overcome using stochastic VI (Hoffman et al., 2013), which subsamples the parameters $\Theta$ inferred for at each optimization step. However, using SVI, the weights for the posterior for a local parameter $\theta \in \Theta$ are only updated when $\theta$ is visited by the algorithm. In the presence of hundreds of thousands of such local parameters, stochastic VI can become prohibitively slow.

In this work, we introduce the concept of *plate amortization* (PAVI) for fast inference in large scale HBMs. Instead of considering the inference over local parameters $\theta$ as separate problems, our main idea is to share both the parameterization and learning across those local parameters –or equivalently across a model's *plates*. We first propose an algorithm to automatically derive an expressive yet parsimoniously-parameterized variational family from a plate-enriched HBM. We then propose a hierarchical stochastic optimization scheme to train this architecture efficiently, obtaining orders of magnitude faster convergence. We detail two variants of our method, with different trade-offs between parameterization and inference quality. PAVI leverages the repeated structure of plate-enriched HBMs via an original combination of amortization and stochastic training. Through this combination, our main claim is to enable inference over arbitrarily large population studies –up to a million RVs– with reduced parameterization and training time as the cardinality of the problem augments. We illustrate this by applying PAVI to a challenging human brain cortex parcellation, featuring inference over a cohort of 1000 subjects with tens of thousands of measurements per subject, demonstrating a significant step towards scalable, expressive and fast VI.

## 2 METHODS

### 2.1 HIERARCHICAL BAYESIAN MODELS (HBMS), TEMPLATES AND PLATES

Our objective is to perform inference in large population studies. As an example of how inference becomes impractical in this context, consider $\mathcal{M}$ in fig. 1 as a model for the height distribution in a population. $\theta_{2,0}$ denotes the mean height across the population. $\theta_{1,0}, \theta_{1,1}, \theta_{1,2}$ denote the mean heights for 3 groups of subjects, distributed around the population mean. $X_0, X_1$ represent the observed heights of 2 subjects from group 0, distributed around the group mean. Given the observed subject heights $X$, the goal is to determine the posterior distributions of the group means $p(\theta_{1,n}|X)$ and population mean $p(\theta_{2,0}|X)$. As the number of groups and subjects per group augments, the parameterization and time required to infer the posteriors of a growing number of RVs become prohibitively large. Our goal is to keep this inference computationally tractable.

In the rest of this section, we define the inference problem and the associated notations. Population studies can be modelled using Hierarchical Bayesian Models (HBMs). Those HBMs feature samples from common conditional distributions at multiple levels. For instance subject heights are i.i.d. given group heights, which are i.i.d. given population height. Due to data being collected across hundreds of subjects (Kong et al., 2019), HBMs representing population studies feature

thousands of RVs, most of which correspond to symmetrical distributions. For instance, the distribution of a subject's height given the group height is the same across all subjects. As a result, those large HBMs $\mathcal{M}$ can be compactly represented via plate-enriched Directed Acyclic Graph (DAG) templates $\mathcal{T}$ (Gilks et al., 1994; Koller & Friedman, 2009). The plates in the template $\mathcal{T}$ translate graphically the hierarchical i.i.d. structure in the model $\mathcal{M}$. We exploit this plate-induced repeated structure to design scalable inference methods.

We denote $\mathcal{T}$'s vertices as $X$ and $\Theta = \{\theta_i\}_{i=1..I}$. $X$ denotes the RVs observed during inference, and $\Theta$ the parameters we infer: our goal is to approximate the posterior distribution $p(\Theta|X)$. We denote $\mathcal{T}$'s plates as $\{\mathcal{P}_p\}_{p=0..P}$, and the plates $\theta_i$ belongs to as $\mathrm{Plates}(\theta_i)$. In the toy example from fig. 1, there are 2 latent RV templates: $\theta_1$ and $\theta_2$, two plates $\mathcal{P}_0, \mathcal{P}_1$ and we want to approximate $p(\theta_1, \theta_2|X)$.

We build upon the interplay between the template $\mathcal{T}$ and the HBM $\mathcal{M}$. To go from one representation to the other, $\mathcal{T}$ can be *grounded* into $\mathcal{M}$ given plate cardinalities $\{\mathrm{Card}(\mathcal{P}_p)\}_{p=0..P}$ (Koller & Friedman, 2009). $\mathrm{Card}(\mathcal{P})$ represents the number of elements in the plate $\mathcal{P}$, for instance the number of groups in the study. The *grounding* operation instantiates the repeated structures symbolized by the plates: a given RV *template* $\theta_i$ corresponds to multiple *ground* RVs $\{\theta_{i,n}\}_{n=0..N_i}$ with the same parametric form, where $N_i = \prod_{\mathcal{P} \in \mathrm{Plates}(\theta_i)} \mathrm{Card}(\mathcal{P})$. Template grounding is illustrated in fig. 1, where $\mathcal{T}$ is instantiated into models $\mathcal{M}$. In PAVI, we consider the inference over the ground RVs $\theta_{i,n}$ as symmetrical problems, over which we share parameterization and learning.

Our goal is to perform inference over the *full* model $\mathcal{M}$, which is associated to the density $p$. In $p$, the plate structure indicates that a RV template $\theta_i$ is associated to a conditional distribution $p_i$ shared across all ground RVs $\theta_{i,n}$:

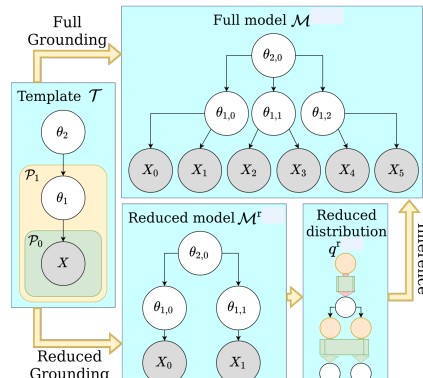

Figure 1: **Plate Amortized Variational Inference (PAVI) working principle.** Starting on the left, the graph template $\mathcal{T}$ is grounded into 2 separate HBMs: $\mathcal{M}$ (top) and $\mathcal{M}^{\mathrm{r}}$ (down) of respective plate cardinalities $(3, 2)$ –large– and $(2, 1)$ –small. Based on $\mathcal{M}^{\mathrm{r}}$, the reduced distribution $q^{\mathrm{r}}$ is constructed. We train $q^{\mathrm{r}}$ over data slices of small cardinality, before performing inference over the full model of large cardinality.

$$\log p(\Theta, X) = \sum_{n=0}^{N_X} \log p_X(x_n|\pi(x_n)) + \sum_{i=1}^{I}\sum_{n=0}^{N_i} \log p_i(\theta_{i,n}|\pi(\theta_{i,n})) \ , \tag{1}$$

where $\pi(\theta_{i,n})$ are the parents of the RV $\theta_{i,n}$ –conditioning its distribution. We denote with a $\bullet_X$ index all variables related to the observed RVs $X$. $\mathcal{M}$ typically features large plate cardinalities $\mathrm{Card}(\mathcal{P})$, and thus many ground RVs, making it computationally intractable. Instead of inferring directly over $\mathcal{M}$, we will train over a smaller replica of $\mathcal{M}$. To this end, we instantiate the template $\mathcal{T}$ into a second HBM $\mathcal{M}^{\mathrm{r}}$ –the *reduced* model– of tractable plate cardinalities $\mathrm{Card}^{\mathrm{r}}(\mathcal{P}) \ll \mathrm{Card}(\mathcal{P})$. $\mathcal{M}^{\mathrm{r}}$ has the same template as $\mathcal{M}$, meaning the same dependency structure and the same parametric form for its distributions. The only difference lies in $\mathcal{M}^{\mathrm{r}}$'s smaller cardinalities, resulting in fewer ground RVs, as visible in fig. 1. We train over the tractable reduced model $\mathcal{M}^{\mathrm{r}}$ to obtain a variational distribution $q$ usable to approximate the posterior for the intractable target model $\mathcal{M}$.

## 2.2 PAVI ARCHITECTURE

### 2.2.1 FULL VARIATIONAL FAMILY DESIGN

Here we define the variational distribution $q$ –corresponding to the full model $\mathcal{M}$– introduced in the previous section. $q$ is the distribution used for inference, but that –due to $\mathcal{M}$'s large cardinalities– we cannot directly train over. To derive $q$, we push forward the prior $p(\Theta)$ using trainable normalizing flows, denoted as $\mathcal{F}$ (Papamakarios et al., 2019). To every ground RV $\theta_{i,n}$, we associate a learnable

flow $\mathcal{F}_{i,n}$ to approximate its posterior distribution:

$$
\begin{aligned}
\log q(\Theta) &= \sum_{i=1}^{I} \sum_{n=0}^{N_i} \log q_{i,n}(\theta_{i,n}|\pi(\theta_{i,n})) \ , \\
\log q_{i,n}(\theta_{i,n}|\pi(\theta_{i,n})) &= -\log \left| \det J_{\mathcal{F}_{i,n}}(u_{i,n}) \right| + \log p_i(u_{i,n}|\pi(\theta_{i,n})) \ , \\
u_{i,n} &= \mathcal{F}_{i,n}^{-1}(\theta_{i,n}) \ ,
\end{aligned}
\tag{2}
$$

where $q_{i,n}$ is the push-forward of the prior $p_i$ through the flow $\mathcal{F}_{i,n}$. This push-forward is illustrated in fig. 2, where flows $\mathcal{F}$ push RVs $u$ into $\theta$. This *cascading* scheme was first introduced by Ambrogioni et al. (2021b), and makes $q$ inherit the conditional dependencies of the prior $p$. More details about the dependencies modelled in the variational distribution can be found in appendix C.5.

### 2.2.2 PLATE AMORTIZATION

Here we introduce plate amortization: sharing the parameterization of density estimators across a model's plates. Traditional VI searches for a distribution $q(\Theta; \phi)$ that best approximates the posterior of $\Theta$ given a value $\mathbf{X}_0$ for $X$: $q(\Theta; \phi_0) \simeq p(\Theta|X = \mathbf{X}_0)$. When presented with a new value $\mathbf{X}_1$, optimization has to be performed again to search for the weights $\phi_1$, such that $q(\Theta; \phi_1) \simeq p(\Theta|X = \mathbf{X}_1)$. *Sample amortized* inference (Zhang et al., 2019; Cremer et al., 2018) instead infers in the general case, regressing the weights $\phi$ using an *encoder* $f$ of the observed data $\mathbf{X}$: $q(\Theta; \phi = f(\mathbf{X})) \simeq p(\Theta|X = \mathbf{X})$. The cost of learning the encoder weights is *amortized* since inference over any new sample $\mathbf{X}$ requires no additional optimization. We propose to exploit the concept of amortization, but to apply it at a different granularity, leading to our notion of *plate amortization*.

Similar to amortizing across the different data samples $\mathbf{X}$, we amortize across the different ground RVs $\{\theta_{i,n}\}_{n=0..N_i}$ corresponding to the same RV template $\theta_i$. Instead of casting every flow $\mathcal{F}_{i,n}$ –defined in eq. (2)– as an separate, fully-parameterized flow, we will share some parameters across the $\mathcal{F}_{i,n}$. To the template $\theta_i$, we associate a conditional flow $\mathcal{F}_i(\ \cdot\ ; \phi_i, \bullet)$ with weights $\phi_i$ shared across all the $\{\theta_{i,n}\}_{n=0..N_i}$. The flow $\mathcal{F}_{i,n}$ associated to a given ground RV $\theta_{i,n}$ will be an instance of this conditional flow, conditioned by an encoding $\mathbf{E}_{i,n}$:

$$
\mathcal{F}_{i,n} = \mathcal{F}_i(\ \cdot\ ; \phi_i, \mathbf{E}_{i,n}) \quad \text{yielding} \quad q_{i,n} = q_{i,n}(\theta_{i,n}|\pi(\theta_{i,n}); \phi_i, \mathbf{E}_{i,n})
\tag{3}
$$

The distributions $q_{i,n}$ thus have 2 sets of weights, $\phi_i$ and $\mathbf{E}_{i,n}$, creating a parameterization trade-off. Concentrating all of $q_{i,n}$'s parameterization into $\phi_i$ results in all the ground RVs $\theta_{i,n}$ having the same posterior distribution. On the contrary, concentrating all of $q_{i,n}$'s parameterization into $\mathbf{E}_{i,n}$ allows the $\theta_{i,n}$ to have completely different posterior distributions. But in a large cardinality setting, this freedom can result in a massive number of weights, proportional to the number of ground RVs times the encoding size. This double parameterization is therefore efficient when the majority of the weights of $q_{i,n}$ is concentrated into $\phi_i$. Using normalizing flows $\mathcal{F}_i$, the burden of approximating the correct parametric form for the posterior is placed onto $\phi_i$, while the $\mathbf{E}_{i,n}$ encode lightweight summary statistics specific to each $\theta_{i,n}$. In section 2.3.2, we also see that this shared parameterization has synergies with stochastic training.

### 2.3 PAVI STOCHASTIC TRAINING

### 2.3.1 REDUCED DISTRIBUTION AND LOSS

Because of the large number of RVs in $\mathcal{M}$, optimizing over the full distribution $q$ –defined in eq. (2)– is computationally intractable. Instead, we optimize over a distribution with the smaller cardinalities of the reduced model. At each optimization step $t$, we randomly choose inside $\mathcal{M}$ paths of reduced cardinality, as visible in fig. 2. Selecting paths is equivalent to selecting from $X$ a subset $X^{\mathfrak{r}}[t]$ of size $N_X^{\mathfrak{r}}$, and from $\Theta$ a subset $\Theta^{\mathfrak{r}}[t]$. For a given $\theta_i$, we denote as $\mathcal{B}_i[t]$ the batch of selected ground RVs, of size $N_i^{\mathfrak{r}}$. Inferring over $\Theta^{\mathfrak{r}}[t]$, we will simulate training over the full distribution $q$:

$$
\log q^{\mathfrak{r}}(\Theta^{\mathfrak{r}}[t]) = \sum_{i=1}^{I} \frac{N_i}{N_i^{\mathfrak{r}}} \sum_{n \in \mathcal{B}_i[t]} \log q_{i,n}(\theta_{i,n}|\pi(\theta_{i,n}))
\tag{4}
$$

where the factor $N_i/N_i^{\mathfrak{r}}$ simulates that we observe as many ground RVs as in $\mathcal{M}$ by repeating the RVs from $\mathcal{M}^{\mathfrak{r}}$ (Hoffman et al., 2013). Similarly, the loss used at step $t$ is the reduced ELBO constructed using $X^{\mathfrak{r}}[t]$ as observed RVs:

$$\log p^{\mathfrak{r}}(X^{\mathfrak{r}}[t], \Theta^{\mathfrak{r}}[t]) = \frac{N_X}{N_X^{\mathfrak{r}}} \sum_{n \in \mathcal{B}_X[t]} \log p_X(x_n|\pi(x_n)) + \sum_{i=1}^{I} \frac{N_i}{N_i^{\mathfrak{r}}} \sum_{n \in \mathcal{B}_i[t]} \log p_i(\theta_{i,n}|\pi(\theta_{i,n})) \tag{5}$$

$$\mathrm{ELBO}^{\mathfrak{r}}[t] = \mathbb{E}_{\Theta^{\mathfrak{r}} \sim q^{\mathfrak{r}}} \left[ \log p^{\mathfrak{r}}(X^{\mathfrak{r}}[t], \Theta^{\mathfrak{r}}[t]) - \log q^{\mathfrak{r}}(\Theta^{\mathfrak{r}}[t]) \right]$$

This scheme can be viewed as the instantiation of $\mathcal{M}^{\mathfrak{r}}$ over batches of $\mathcal{M}$'s ground RVs. In fig. 2 we see that $q^{\mathfrak{r}}$ has the cardinalities of $\mathcal{M}^{\mathfrak{r}}$, and replicates its conditional dependencies. This training is analogous to stochastic VI (Hoffman et al., 2013), generalized with multiple hierarchies and minibatches of RVs. Our novelty lies in the interaction of this stochastic scheme with plate amortization, as explained in the next section.

### 2.3.2 SHARING LEARNING ACROSS PLATES

Here we detail how our shared parameterization –detailed in section 2.2.2– combined with our stochastic training scheme, results in faster inference. In traditional stochastic VI, every $\theta_{i,n}$ corresponding to the same template $\theta_i$ is associated to individual weights. Those weights are trained only when $\theta_{i,n}$ is visited by the algorithm, that is to say at step $t$ when $n \in \mathcal{B}_i[t]$. As plates become larger, this event becomes rare. If $\theta_{i,n}$ is furthermore associated to a highly-parameterized density estimator –such as a normalizing flow– many optimization steps are required for $q_{i,n}$ to converge. The combination of those two items leads to slow training.

Instead, our idea is to share the learning across the ground RVs $\theta_{i,n}$. Due to the problem's plate structure, we consider the inference over the $\theta_{i,n}$ as different instances of a common density estimation task. In PAVI, a large part of the parameterization of the estimators $q_{i,n}(\theta_{i,n}|\pi(\theta_{i,n}); \phi_i, \mathbf{E}_{i,n})$ is

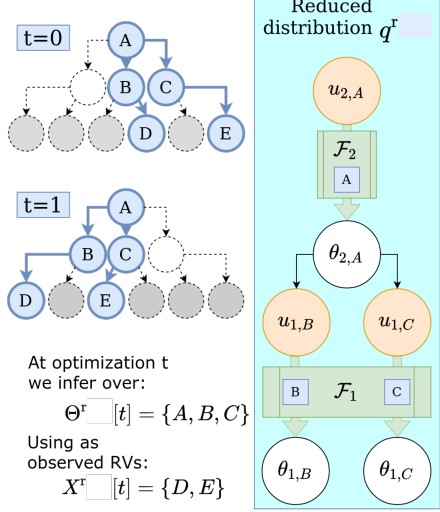

Figure 2: **PAVI stochastic training scheme** The reduced distribution $q^{\mathfrak{r}}$ features 2 conditional normalizing flows $\mathcal{F}_1$ and $\mathcal{F}_2$ respectively associated to the RV templates $\theta_1$ and $\theta_2$. During the stochastic training, $q^{\mathfrak{r}}$ is instantiated over different branchings of the full model $\mathcal{M}$ –highlighted in blue on the left. The branchings have the cardinalities of $\mathcal{M}^{\mathfrak{r}}$ and change at each stochastic training step $t$. The branching determine the encodings $\mathbf{E}$ conditioning the flows $\mathcal{F}$ –as symbolised by the letters A, B, C– and the observed data slice –as symbolised by the letters D, E.

mutualized via the plate-wide-shared weights $\phi_i$. This means that most of the weights of the flows $\mathcal{F}_{i,n}$ –concentrated in $\phi_i$– are trained at every optimization step, across all the selected batches $\mathcal{B}_i[t]$. This results in drastically faster convergence compared to SVI –as seen in exp. 3.1.

### 2.4 ENCODING SCHEMES

PAVI shares the parameterization and learning of density estimators across an HBM's plates. In practice the distributions $q_{i,n}(\theta_{i,n}|\pi(\theta_{i,n}); \phi_i, \mathbf{E}_{i,n})$ –defined in eq. (3)– with different $n$ only differ through the value of the encodings $\mathbf{E}_{i,n}$. We detail one core and one alternate scheme to derive those encodings:

**Free plate encodings (PAVI-F)** In our core implementation, $\mathbf{E}_{i,n}$ are free weights. We define encoding arrays with the cardinality of the full model $\mathcal{M}$, one array $\mathbf{E}_i = [\mathbf{E}_{i,n}]_{n=0..N}$ per template $\theta_i$. This means that an additional ground RV –for instance adding a subject in a population study– requires an additional encoding vector. The associated increment in the total number of weights is much lighter than the addition of a fully parameterized normalizing flow –as would be the case in the non-plate-amortized regime. The PAVI-F scheme cannot be sample amortized: when presented with an unseen $\mathbf{X}$, though $\phi_i$ can be kept as an efficient warm start, the optimal values for the encodings $\mathbf{E}_{i,n}$ have to be searched again. During training, the encodings $\mathbf{E}_{i,n}$ corresponding to

$n \in \mathcal{B}_i[t]$ are sliced from the arrays $\mathbf{E}_i$ and are optimized for along with $\phi_i$. In the toy example from fig. 2, at $t = 0$, $\mathcal{B}_1[0] = \{1, 2\}$ and the trained encodings are $\{\mathbf{E}_{1,1}, \mathbf{E}_{1,2}\}$, and at $t = 1$ $\mathcal{B}_1[1] = \{0, 1\}$ and we train $\{\mathbf{E}_{1,0}, \mathbf{E}_{1,1}\}$. In appendix A.2, we show that the training of PAVI-F is unbiased: training over $\mathcal{M}^{\mathfrak{r}}$, we converge to the same distribution as if training directly over $\mathcal{M}$.

**Deep set encoder (PAVI-E)** The parameterization of PAVI-F scales lightly but linearly with $\mathrm{Card}(\mathcal{P})$. This scaling could become unaffordable in large population studies. We thus propose an alternate scheme –PAVI-E– with a parameterization independent of cardinalities. In this more experimental scheme, free encodings are replaced by an encoder $f$ with weights $\eta$ applied to the observed data: $\mathbf{E} = f(\mathbf{X}; \eta)$. As encoder $f$ we use a *deep-set* architecture exploiting the data's plate-induced permutation invariance –detailed in appendix A.1.3 (Zaheer et al., 2018; Lee et al., 2019). This scheme allows for *sample amortization* across different data samples $\mathbf{X}_0, \mathbf{X}_1, \dots$. Note that an encoder will be used to generate the encodings whether inference is sample amortized or not. During training, shared learning is further amplified as all the architecture's weights $-\phi_i$ and $\eta-$ are trained at every step $t$. To collect the encodings to plug into $q^{\mathfrak{r}}$, we build up on a property of $f$: *set size generalization* (Zaheer et al., 2018). Instead of encoding the full-sized data $\mathbf{X}$, $f$ is applied to the slice $\mathbf{X}^{\mathfrak{r}}[t]$. This amounts to aggregating summary statistics across a subset of the observed data (Lee et al., 2019; Agrawal & Domke, 2021). This property is intensified in the sample amortized context: we train a sample amortized family over the lightweight model $\mathcal{M}^{\mathfrak{r}}$, and use it "for free" over the heavyweight model $\mathcal{M}$. As detailed in appendix A.2, we rely on a computationally efficient but theoretically biased training scheme for PAVI-E. The negative impact of this bias on PAVI-E's performance was seldom noticeable and always marginal throughout our experiments.

**Summary** In section 2.2 we derived an architecture sharing its parameterization across a model's plates. In section 2.3 we derived a stochastic scheme to train this architecture over batches of data. Our novelty lies in this original combination of amortization and stochastic training, which results in significantly faster inference, as demonstrated in the following experiments.

# 3 RESULTS AND DISCUSSION

Throughout this section we use the ELBO as a proxy to the KL divergence between the variational posterior and the unknown true posterior. ELBO is measured across 20 samples $\mathbf{X}$, 5 random seeds per sample. The ELBO allows to compare the relative performance of different architectures on a given inference problem. In appendix B.1 we provide with sanity checks to assess the quality of the results. In appendix B.2 we evaluate the impact of the reduced model cardinalities on performance. In appendix B.3 we compare numerically our method against baselines over a Gaussian mixture model; a model featuring the aggregation of higher order summary statistics; and a smaller version of our Neuroimaging model used in section 3.4.

Experimentally, we found the PAVI-F scheme overall faster to train and yielding better inference quality than the PAVI-E scheme. When its parameterization is affordable, PAVI-F should be preferred. PAVI-E nevertheless opens up promising research directions, with the potential for parameterization-constant, time-constant sample-amortized inference as cardinality augments. Though degraded compared to PAVI-F's, PAVI-E's performance is still on par or beats baselines in a variety of inference tasks –see exp. 3.3&B.3.

## 3.1 PLATE AMORTIZATION AND CONVERGENCE SPEED

In this experiment, we illustrate how plate amortization results in faster training. We consider the following Gaussian Random Effects model (GRE):

$$
\begin{aligned}
X_{n_1, n_0} | \theta_{1, n_1} &\sim \mathcal{N}(\theta_{1, n_1}, \sigma_x^2) \quad \begin{smallmatrix} \forall n_1 = 1..\,\mathrm{Card}(\mathcal{P}_1) \\ \forall n_0 = 1..\,\mathrm{Card}(\mathcal{P}_0) \end{smallmatrix} \\
\theta_{1, n_1} | \theta_{2, 0} &\sim \mathcal{N}(\theta_{2, 0}, \sigma_1^2) \quad \forall n_1 = 1..\,\mathrm{Card}(\mathcal{P}_1) \qquad \theta_{2, 0} \sim \mathcal{N}(\vec{0}_D, \sigma_2^2) \ ,
\end{aligned}
\tag{6}
$$

where $D$ represents the data $\mathbf{X}$'s feature size, with group means $\theta_1$ and population means $\theta_2$ as D-dimensional Gaussians. We opt in this equation for a double indexing scheme instead of a simple indexing as in our methods. The GRE model features two nested plates: the group plate $\mathcal{P}_1$ and the sample plate $\mathcal{P}_0$ as in fig. 1. Inferring over the GRE model, the objective is to retrieve the posterior distribution of the group and population means given the observed sample.

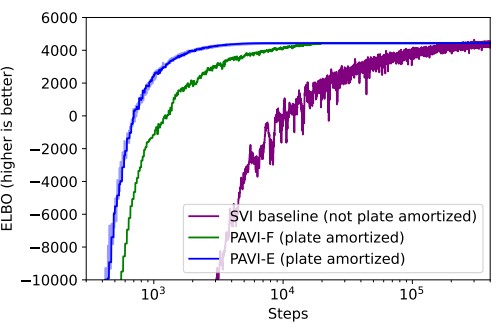 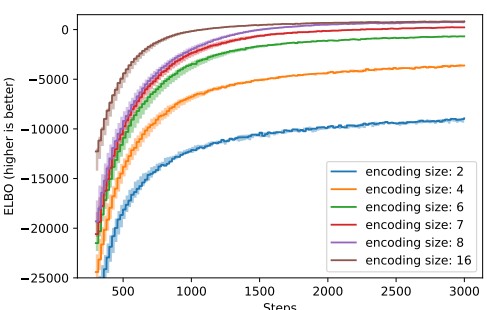

Figure 3: **Left panel: Plate amortization increases convergence speed** Plot of the ELBO (higher is better) as a function of the optimization steps (log-scale) for our methods PAVI-F (in green) and PAVI-E (in blue) versus a non-plate-amortized baseline (in purple). Due to plate amortization, our method converges ten to a hundred times faster to the same asymptotic ELBO as its non-plate-amortized counterpart.; **Right panel: Encodings as ground RVs summary statistics** Plot of the ELBO (higher is better) as a function of the optimization steps for the PAVI-F architecture with increasing encoding sizes. As the encoding size augments, so does the asymptotic performance, until reaching the dimensionality of the posterior's sufficient statistics ($D = 8$), after which performance plateaus. Encoding size allows for a clear trade-off between memory and inference quality.

Here we set $D = 8$, $\mathrm{Card}(\mathcal{P}_1) = 100$ and $\mathrm{Card}^{\mathrm{r}}(\mathcal{P}_1) = 2$. We compare our PAVI architecture to a stochastic non-plate-amortized baseline with the same architecture as PAVI (Hoffman et al., 2013). The only difference is that ground RVs $\theta_{i,n}$ are associated in the baseline to individual fully-parameterized flows $\mathcal{F}_{i,n}$ instead of sharing the same conditional flow $\mathcal{F}_i$ –as described in section 2.2.2. Figure 3 (left) displays the evolution of the ELBO for the baseline and PAVI with free encoding (PAVI-F) and deep set encoders (PAVI-E). We see that both plate amortized methods reach asymptotic ELBO equal to the non-plate-amortized baseline's, but with orders of magnitudes faster convergence, and more numerical stability. This stems from the individual flows $\mathcal{F}_{i,n}$ in the baseline only being trained when the corresponding $\theta_{i,n}$ is visited by the stochastic training, while the shared flow $\mathcal{F}_i$ is updated at every optimization step in PAVI. We also note that the PAVI-E scheme converges faster in theory than PAVI-F –in terms of number of steps– by sharing not only the training of the flows, but also of the encoder through the optimization. In practice however, computing the encodings results in longer steps, and in slower convergence, as illustrated in section 3.3.

## 3.2 IMPACT OF ENCODING SIZE

Here we illustrate the role of encodings as ground RV posterior's summary statistics –as described in section 2.2.2. We use the GRE HBM detailed in eq. (6), using $D = 8$, $\mathrm{Card}(\mathcal{P}_1) = 20$ and $\mathrm{Card}^{\mathrm{r}}(\mathcal{P}_1) = 2$. We use a single PAVI-F architecture, varying the size of the encodings $\mathbf{E}_{i,n}$ –see section 2.4. Due to plate amortization, encodings determines how much individual information each RV $\theta_{i,n}$ is associated to. The encoding size –varying from 2 to 16– is to be compared with the dimensionality of the problem, $D = 8$. In GRE, $D = 8$ corresponds to the size of the sufficient statistics needed to reconstruct the posterior of a group mean –all other statistics such as the variance being shared between the group means. Figure 3 (right) shows how the asymptotic performance steadily increases when the encoding size augments, and plateaus once reaching the sufficient summary statistic size $D = 8$. Interestingly, increasing the encoding size also leads to faster convergence: redundancy can likely be exploited in the optimization. Encoding size appears as a unequivocal hyperparameter allowing to trade inference quality for computational efficiency. Increasing the encoding size also leads experimentally to diminishing returns in terms of performance. This property can be exploited in large settings to drastically reduce the memory footprint of inference while maintaining acceptable performance –choosing the encoding size approximately equal to the expected size of the sufficient statistics.

## 3.3 SCALING WITH PLATE CARDINALITIES

Here we put in perspective the gains from plate amortization when scaling up an inference problem's cardinality. We consider the GRE model in eq. (6) with $D = 2$ and augment the plate cardinalities

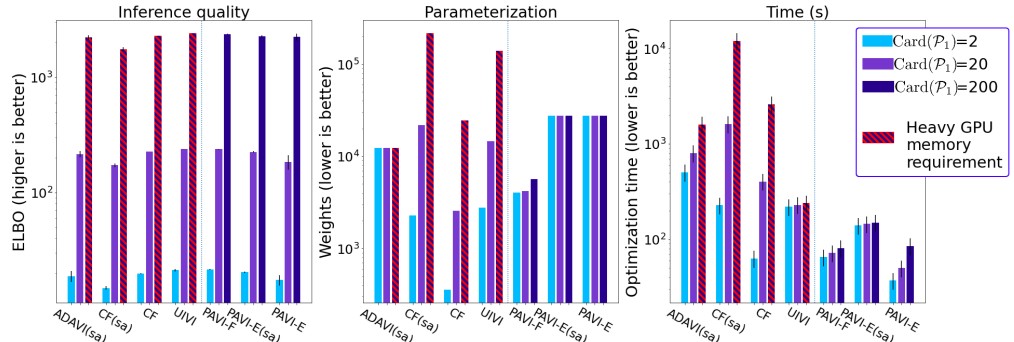

Figure 4: **PAVI provides with favorable parameterization and training time as the cardinality of the target model augments** Our architecture PAVI is displayed on the right of each panel. We augment the cardinality $\mathrm{Card}(\mathcal{P}_1)$ of the GRE model –described in eq. (6). While doing so, we compare 3 different metrics: *In the first panel:* inference quality, as measured by the ELBO. None of the presented SOTA architecture's performance degrades as the cardinality of the problem augments. *In the second pannel:* parameterization, comparing the number of trainable weights of each architecture. PAVI –similar to ADAVI– displays a constant number of weights as the cardinality of the problem increases –or almost constant for PAVI-F. *Third panel:* GPU training time. Benefiting from learning across plates, PAVI has a short and almost constant training time as the cardinality of the problem augments. At $\mathrm{Card}(\mathcal{P}_1) = 200$, CF, UIVI and ADAVI required large GPU memory, a constraint absent from PAVI due to its stochastic training.

$(\mathrm{Card}(\mathcal{P}_1), \mathrm{Card}^{\mathtt{r}}(\mathcal{P}_1)) : (2, 1) \rightarrow (20, 5) \rightarrow (200, 20)$. In doing so, we augment the number of parameters $\Theta : 6 \rightarrow 42 \rightarrow 402$.

**Baselines** We compare our PAVI architecture against 3 state-of-the-art baselines. Cascading Flows (CF) (Ambrogioni et al., 2021b) is a non-plate-amortized structured VI architecture improving on the baseline presented in section 3.1. CF push the prior $p$ into the posterior $q$ using *Highway Flows*. CF follows a cascading dependency structure, complemented with a backward auxiliary coupling. ADAVI (Rouillard & Wassermann, 2022) is a structured VI architecture similar to PAVI, with constant parameterization with respect to a problem's cardinality, but large training times and memory. ADAVI has several limitations compared to PAVI: ADAVI implements a Mean Field approximation (Blei et al., 2017) while PAVI implements a cascading flow; ADAVI is limited pyramidal HBMs while PAVI tackles generic plate-enriched HBMs; ADAVI is limited to a full-model sample-amortized variant. Unbiased Implicit VI (UIVI) is an unstructured implicit VI architecture. UIVI infers over the full parameter space –without any SVI-amenable factorization– by reparameterizing a base distribution with a stochastic transform. For all architectures, we indicate with the suffix *(sa) sample amortization*, corresponding to the classical meaning of amortization, as detailed in section 2.2.2. More implementation details can be found in appendix B.4.

As the cardinality of the problem augments, fig. 4 shows how PAVI maintains a state-of-the-art inference quality, while being more computationally attractive. Specifically, in terms of parameterization, both ADAVI and PAVI-E provide with a heavyweight but constant parameterization as the cardinality $\mathrm{Card}(\mathcal{P}_1)$ of the problem augments. Comparatively, both CF and PAVI-F's parameterization scale linearly with $\mathrm{Card}(\mathcal{P}_1)$, but with a drastically lighter augmentation for PAVI-F. For an additional ground RV, CF requires an additional fully parameterized normalizing flow, whereas PAVI-F only requires an additional lightweight encoding vector. UIVI's parameterization scales quadratically with the size of the parameter space $\Theta$, due to a neural network regressing a transform applied to a base distribution with the size of $\Theta$. In detail, PAVI-F's parameterization due to the plate-wide-shared $\phi_1$ represents a constant $\approx 2k$ weights, while the part due to the encodings $\mathbf{E}_{1,n}$ grows linearly from 16 to 160 to $1.6k$ weights. Note that PAVI's stochastic training also allows for a controlled GPU memory during optimization, removing the need for a larger memory as the cardinality of the problem augments –a hardware constraint that can become unaffordable at very large cardinalities. To remove this memory constraint, CF could be trained stochastically, but –without plate amortization– would suffer from slower inference, as illustrated in section 3.1. In contrast, UIVI could not be trained stochastically, as it infers over the full $\Theta$ at once instead of factorizing it. As a result UIVI would be ultimately limited by memory to

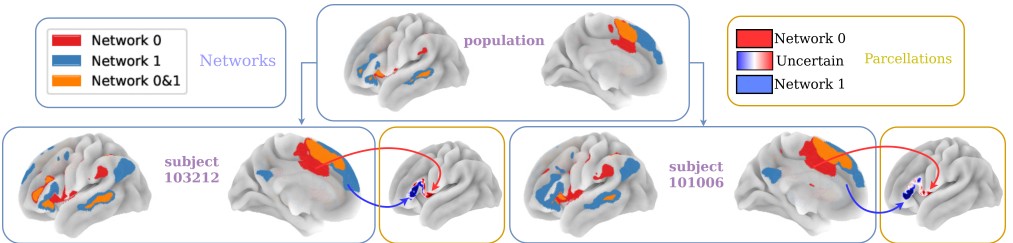

Figure 5: **Probabilistic parcellation of Brocas's area** PAVI can be applied in the challenging context of Neuroimaging population studies. For a cohort of 1000 subjects, 2 of which are represented here –in the bottom 2 items– we present 2 results. First, *connectivity networks* with the brain's left hemisphere –left purple items: those represent the brain regions vertices with each label are "wired" to. Second, Broca's area probabilistic *parcellation* –rightmost orange items: we cluster the brain's vertices, associating them to *connectivity networks*. Our method models uncertainty: coloring transitions from red to an uncertain white to blue, representing the probability of a vertex to belong to one network or the other.

infer over larger problems. In terms of convergence speed, PAVI benefits from plate amortization to have orders of magnitude faster convergence compared to structured VI baselines CF and ADAVI. For UIVI, as the cardinality augments, training amounts to evaluating a neural network with increasingly larger layers. GPU training time is therefore constant, but this property would not translate to larger problems as the GPU memory would become insufficient. Plate amortization is particularly significant for the PAVI-E(sa) scheme, in which a sample-amortized variational family is trained over a dataset of reduced cardinality, yet performs "for free" inference over a HBM of large cardinality. Maintaining $\text{Card}^{\text{r}}(\mathcal{P}_1)$ constant while $\text{Card}(\mathcal{P}_1)$ augments allows for a constant parameterization *and training time* as the cardinality of the problem augments –without any maximum cardinality, contrary to UIVI. The effect of plate amortization is particularly noticeable at $\text{Card}(\mathcal{P}_1) = 200$ between the PAVI(sa) and CF(sa) architectures, where PAVI performs amortized inference with $10\times$ fewer weights and $100\times$ lower training time. Scaling even higher the cardinality of the problem –$\text{Card}(\mathcal{P}_1) = 2000$ for instance– renders ADAVI, CF and UIVI computationally intractable, while PAVI maintains a light memory footprint, and a short training time, as exemplified in the next experiment.

### 3.4 NEUROIMAGING APPLICATION: BROCA'S AREA PARCELLATION

To illustrate its usefulness, we apply PAVI to a challenging population study for Broca's area's functional *parcellation*. A *parcellation* clusters brain vertices into different *connectivity networks*: labels describing co-activation with the rest of the brain –as measured using functional Magnetic Resonance Imaging (fMRI). Different subjects exhibit a strong variability, as in fig. 5. However, fMRI is costly to acquire: few noisy data is usually gathered for a given subject. It is thus essential to combine information across subjects and to display uncertainty in the results. Those 2 points motivate Hierarchical Bayesian Modelling and VI in Neuroimaging (Kong et al., 2019): we search the posteriors of connectivity networks and vertex labels, measuring fMRI over a large cohort of subjects. We use the HCP dataset (Van Essen et al., 2012): 2 acquisitions from a 1000 subjects, with thousands of measures per acquisition, for over a million parameters $\Theta$. We use a model with 3 plates: subjects, sessions and brain vertices. None of the baselines presented in section 3.3 –CF, ADAVI, UIVI– can computationally tackle this high cardinality problem. We nevertheless show superior performance over those baselines over a tractable problem size with 2 thousand parameters in appendix B.3.3. Despite the massive dimensionality of the problem, thanks to plate amortization PAVI converges in a dozen epochs, under an hour of GPU time. Results are visible in fig. 5, supporting the hypothesis of a functional bi-partition of Broca's area into a posterior part involved in phonology and an anterior part in lexical/semantic processing –following the anatomy of *pars opercularis* and *triangularis* (Heim et al., 2009; Zhang et al., 2020).

**Conclusion** In this work we present the novel PAVI architecture, combining a structured variational family and a stochastic training scheme. PAVI is based the concept of plate amortization, allowing to share parameterization and learning across a model's plates. We demonstrated the positive impact of plate amortization on training speed and scaling to large plate cardinality regimes, making a significant step towards scalable, expressive Variational Inference.

REPRODUCIBILITY STATEMENT

All experiments were performed in Python using the Tensorflow Probability library (Dillon et al., 2017). All experiments were conducted on computational cluster nodes equipped with a Tesla V100-16Gb GPU and 4 AMD EPYC 7742 64-Core processors. VRAM intensive experiments in fig. 4 were performed on an Ampere 100 PCIE-40Gb GPU. Appendix B.4 lists implementation details for all our synthetic experiments. Appendix B.5 is related to our Neuroimaging experiment 3.4, and details both our data pre-processing steps and our implementation. As part of our submission we furthermore packaged and release the code associated to our experiments.

ETHICS STATEMENT

Our Neuroimaging data come from the Human Connectome Project dataset (Van Essen et al., 2012). All data in the HCP is strongly anonymized, as per the HCP protocols. We used in this paper only Open Access imaging data data, following the HCP data use terms.

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

## SUPPLEMENTAL MATERIAL

## A  SUPPLEMENTAL METHODS

### A.1  PAVI IMPLEMENTATION DETAILS

#### A.1.1  PLATE BRANCHINGS AND STOCHASTIC TRAINING

As exposed in section 2.3.1, at each optimization step $t$ we randomly select branchings inside the full model $\mathcal{M}$, branchings over which we *instantiate* the reduced model $\mathcal{M}^{\mathfrak{r}}$. In doing so, we define batches $\mathcal{B}_i[t]$ for the RV templates $\theta_i$. Those batches have to be coherent with one another: they have to respect the conditional dependencies of the original model $\mathcal{M}$. As an example, if a ground RV is selected as part of $\mathcal{M}^{\mathfrak{r}}$, then its parent RV needs to be selected as well. To ensure this, during the stochastic training we do not sample RVs directly but plates:

1. For every plate $\mathcal{P}_p$, we sample without replacement $\mathrm{Card}^{\mathfrak{r}}(\mathcal{P}_p)$ indices amongst the $\mathrm{Card}(\mathcal{P}_p)$ possible indices.
2. Then, for every RV template $\theta_i$, we select the ground RVs $\theta_{i,n}$ corresponding to the sampled indices for the plates $\mathrm{Plates}(\theta_i)$.
3. The selected ground RVs $\theta_{i,n}$ constitute the set $\Theta^{\mathfrak{r}}[t]$ of parameters appearing in eq. (4). The same procedure yields the observed RV subset $X^{\mathfrak{r}}[t]$ and the data slice $\mathbf{X}^{\mathfrak{r}}[t]$.

For instance, in the toy example from fig. 1, $X_2$ will be chosen iff the index 1 is selected as part of sub-sampling $\mathcal{P}_1$ and the index 0 is selected as part of sub-sampling $\mathcal{P}_0$. Less formally, this is equivalent to going *middle*, then *left* in the full graph representing $\mathcal{M}$. This stochastic choice is illustrated in fig. 2 at $t = 1$ where $X_2$ corresponds to the node $E$. This stochastic strategy also applies to the selected encoding scheme –described in section 2.4&2.3.2– as detailed in the next sections.

#### A.1.2  PAVI-F DETAILS

In section 2.4 we refer to encodings $\mathbf{E}_i = [\mathbf{E}_{i,n}]_{n=0..N}$ corresponding to RV templates $\theta_i$. In practice, we have some amount of sharing for those encodings: instead of defining separate encodings for every RV template, we define encodings for every *plate level*. A plate level is a combination of plates with at least one parameter RV template $\theta_i$ belonging to it:

$$\mathrm{PlateLevels} = \{(\mathcal{P}_k..\mathcal{P}_l) = \mathrm{Plates}(\theta_i)\}_{\theta_i \in \Theta} \tag{A.1}$$

For every plate level, we construct a large encoding array with the cardinalities of the full model $\mathcal{M}$:

$$\mathrm{Encodings} = \{(\mathcal{P}_k..\mathcal{P}_l) \mapsto \mathbb{R}^{\mathrm{Card}(\mathcal{P}_k)\times..\times\mathrm{Card}(\mathcal{P}_l)\times D}\}_{(\mathcal{P}_k..\mathcal{P}_l)\in\mathrm{PlateLevels}}$$
$$\mathbf{E}_i = \mathrm{Encodings}(\mathrm{Plates}(\theta_i)) \tag{A.2}$$

Where $D$ is an encoding size that we kept constant to de-clutter the notation but can vary between plate levels. The encodings for a given ground RV $\theta_{i,n}$ then correspond to an element from the encoding array $\mathbf{E}_i$.

#### A.1.3  PAVI-E DETAILS

In the PAVI-E scheme, encodings are not free weights but the output of en encoder $f(\cdot, \eta)$ applied to the observed data $\mathbf{X}$. In this section we detail the design of this encoder.

As in the previous section, the role of the encoder will be to produce one encoding per plate level. We start from a dependency structure for the plate levels:

$$\forall(\mathcal{P}_a..\mathcal{P}_b) \in \mathrm{PlateLevels} \ ,$$
$$\forall(\mathcal{P}_c..\mathcal{P}_d) \in \mathrm{PlateLevels} \ ,$$
$$(\mathcal{P}_a..\mathcal{P}_b) \in \pi((\mathcal{P}_c..\mathcal{P}_d)) \Leftrightarrow \substack{\exists\theta_i / \mathrm{Plates}(\theta_i)=(\mathcal{P}_a..\mathcal{P}_b) \\ \exists\theta_j / \mathrm{Plates}(\theta_j)=(\mathcal{P}_c..\mathcal{P}_d)} / \theta_j \in \pi(\theta_i) \tag{A.3}$$

note that this dependency structure is in the *backward* direction: a plate level will be the parent of another plate level, if the former contains a RV who has a child in the latter. We therefore obtain

a plate level dependency structure that *reverts* the conditional dependency structure of the graph template $\mathcal{T}$. To avoid redundant paths in this dependency structure, we take the maximum branching of the obtained graph.

Given the plate level dependency structure, we will recursively construct the encodings, starting from the observed data:

$$\forall x \in X: \quad \text{Encodings}(\text{Plates}(x)) = \rho(\mathbf{x}) \tag{A.4}$$

where $\mathbf{x}$ is the observed data for the RV $x$, and $\rho$ is a simple encoder that processes every observed ground RV's value independently through an identical multi-layer perceptron. Then, until we have exhausted all plate levels, we process existing encodings to produce new encodings:

$$\forall (\mathcal{P}_k..\mathcal{P}_l) \in \text{PlateLevels} \; / \; \nexists x \in X, \text{Plates}(x) = (\mathcal{P}_k..\mathcal{P}_l):$$
$$\text{Encodings}((\mathcal{P}_k..\mathcal{P}_l)) = g(\text{Encodings}(\pi(\mathcal{P}_k..\mathcal{P}_l))) \tag{A.5}$$

where $g$ is the composition of attention-based deep-set networks called *Set Transformers* (Lee et al., 2019; Zaheer et al., 2018). For every plate $\mathcal{P}_p$ present in the parent plate level but absent in the child plate level, $g$ will compute summary statistics *across* that plate, effectively contracting the corresponding batch dimensionality in the parent encoding (Rouillard & Wassermann, 2022).

In the case of multiple observed RVs, we run this "backward pass" independently for each observed data –with one encoder per observed RV. We then concatenate the resulting encodings corresponding to the same plate level.

For more precise implementation details, we invite the reader to consult the codebase released with this supplemental material.

## A.2 ANALYSIS OF BIAS IN THE STOCHASTIC TRAINING

A key concern in our stochastic training scheme is its unbiasedness: we want our stochastic optimization to converge to the same variational posterior as if we trained over the full model directly –without any stochasticity. In this section we first show that the PAVI-F is unbiased. Second, we identify strategies to obtain an unbiased PAVI-E scheme, yet show how the approximations we do in practice can theoretically result in biased training. As an important note, the negative impact of this bias on the performance PAVI-E remained limited throughout our experiments –as seen in experiments 3.3, B.2, B.1 and B.3.

### A.2.1 GENERAL DERIVATION (APPLICABLE TO BOTH THE PAVI-F AND PAVI-E SCHEMES)

We first formalize the *plate sampling* strategy described in appendix A.1.1. To every plate $\mathcal{P}$ we associate the RV $I_{\mathcal{P}}$ corresponding to the $\text{Card}^{\mathfrak{r}}(\mathcal{P})$-sized set of indices sampled without replacement from the $\text{Card}(\mathcal{P})$ possible index values. As an example, with a plate $\mathcal{P}_0$ with $\text{Card}(\mathcal{P}_0) = 4$ and $\text{Card}^{\mathfrak{r}}(\mathcal{P}_0) = 2$, $\{0, 2\}$ or $\{2, 3\}$ can be 2 different samples from $I_{\mathcal{P}_0}$. At a given optimization step $t$, we sample independently from the RVs $\{I_{\mathcal{P}_p}\}_{p=0..P}$. This defines the batches $\mathcal{B}_i[t]$ and the distribution $q^{\mathfrak{r}}$ in eq. (4).

To check the unbiasedness of our stochastic training, we need to show that:

$$\mathbb{E}_{I_{\mathcal{P}_0}} \ldots \mathbb{E}_{I_{\mathcal{P}_P}} [\text{ELBO}^{\mathfrak{r}}[t]] = \text{ELBO} \tag{A.6}$$

Where:

$$\text{ELBO} = \mathbb{E}_{\Theta \sim q} [\log p(X, \Theta) - \log q(\Theta)] \tag{A.7}$$

And $\text{ELBO}^{\mathfrak{r}}[t]$ is defined in eq. (5). In that expression, $q$ and $p$ have symmetrical roles. As the ELBO amounts to the difference between the logarithms of distributions $p$ and $q$, we can prove the equality in eq. (A.6) if we prove that the expectation of each reduced distribution is equal to the corresponding full distribution. To prove the equality in eq. (A.6), a sufficient condition is therefore to prove that:

$$\mathbb{E}_{I_{\mathcal{P}_0}} \ldots \mathbb{E}_{I_{\mathcal{P}_P}} [\log q^{\mathfrak{r}}(\Theta^{\mathfrak{r}}[t])] = \mathbb{E}_{I_{\mathcal{P}}} [\log q^{\mathfrak{r}}(\Theta^{\mathfrak{r}}[t])] = \log q(\Theta) \tag{A.8}$$

where to de-clutter the notations we denote the expectation over the collection of RVs $\{I_{\mathcal{P}_p}\}_{p=0..P}$ as $\mathbb{E}_{I_{\mathcal{P}}}$.

Consider a given ground RV $\theta_{i,n}$ corresponding to the RV template $\theta_i$ and to the plates $\text{Plates}(\theta_i)$. At a given stochastic step $t$, $\theta_{i,n}$ will be chosen if and only if its corresponding *branching* is chosen. Recall that when sampling equiprobably without replacement a set of $k$ elements from a population of $n$ elements, a given element will be present in the set with probability $k/n$. We can apply this reasoning to the choice of *branching* corresponding to a given ground RV. For instance, in fig. 1, $X_2$ will be chosen iff the index 1 is selected as part of sub-sampling $\mathcal{P}_1$ and the index 0 is selected as part of sub-sampling $\mathcal{P}_0$. As $\text{Card}^{\mathfrak{r}}(\mathcal{P}_1) = 2$ indices are chosen inside the plate $\mathcal{P}_1$ of full cardinality 3, and $\text{Card}^{\mathfrak{r}}(\mathcal{P}_0) = 1$ indices are chosen inside the plate $\mathcal{P}_0$ of full cardinality 2, $X_2$ is therefore chosen with probability $2/3 \times 1/2$. More formally, for a ground RV $\theta_{i,n}$ we have:

$$\forall n = 0..N_i \ : \ \mathbb{P}(\theta_{i,n} \in \mathcal{B}_i[t]) = \prod_{\mathcal{P} \in \text{Plates}(\theta_i)} \frac{\text{Card}^{\mathfrak{r}}(\mathcal{P})}{\text{Card}(\mathcal{P})}$$
$$= \frac{N_i^{\mathfrak{r}}}{N_i} \tag{A.9}$$

Applying this reasoning to every RV template $\theta_i$, we have that:

$$\mathbb{E}_{I_{\mathcal{P}}} \left[ \log q^{\mathfrak{r}}(\Theta^{\mathfrak{r}}[t]) \right] = \sum_{i=1}^{I} \frac{N_i}{N_i^{\mathfrak{r}}} \mathbb{E}_{I_{\mathcal{P}}} \left[ \sum_{n \in \mathcal{B}_i[t]} \log q_{i,n}(\theta_{i,n} | \pi(\theta_{i,n})) \right]$$
$$= \sum_{i=1}^{I} \frac{N_i}{N_i^{\mathfrak{r}}} \mathbb{E}_{I_{\mathcal{P}}} \left[ \sum_{n=0}^{N_i} \mathbb{1}_{n \in \mathcal{B}_i[t]} \log q_{i,n}(\theta_{i,n} | \pi(\theta_{i,n})) \right] \tag{A.10}$$
$$= \sum_{i=1}^{I} \frac{N_i}{N_i^{\mathfrak{r}}} \sum_{n=0}^{N_i} \mathbb{E}_{I_{\mathcal{P}}} \left[ \mathbb{1}_{n \in \mathcal{B}_i[t]} \log q_{i,n}(\theta_{i,n} | \pi(\theta_{i,n}); \phi_i, \mathbf{E}_{i,n}) \right]$$

where we exploited the fact that the expectation of the sum of RVs is the sum of the expectations, even in the case of dependent RVs. The term $\mathbb{1}_{n \in \mathcal{B}_i[t]} \times \log q_{i,n}(\theta_{i,n} | \pi(\theta_{i,n}); \phi_i, \mathbf{E}_{i,n})$ is the product of 2 RVs –related to the stochastic choice of plate indices:

- the RV $\mathbb{1}_{n \in \mathcal{B}_i[t]}$ is an indicator that $\theta_{i,n}$'s *branching* has been chosen via the stochastic sampling of plate indices. By construction, this RV depends only on the indices of the plates $\mathcal{P} \in \text{Plates}(\theta_i)$.

- the RV $\log q_{i,n}(\theta_{i,n} | \pi(\theta_{i,n}); \phi_i, \mathbf{E}_{i,n})$ depends on $\mathbf{E}_{i,n}$, whose construction depends on the encoding scheme:

  - In the PAVI-F scheme, $\mathbf{E}_{i,n}$ is a constant.

  - In the PAVI-E scheme, $\mathbf{E}_{i,n}$ results of the application of an encoder to the observed data of a subset of $\theta_{i,n}$'s descendants. By construction, this subset will only depend on the indices of plates containing $\theta_i$'s descendants, but not containing $\theta_i$. The value of $\mathbf{E}_{i,n}$ therefore only depends on the indices of plates $\mathcal{P} \notin \text{Plates}(\theta_i)$

As an example of this reasoning, consider the model $\mathcal{M}$ illustrated in fig. 2. We can evaluate both terms for the ground RV $\theta_{1,2}$ in the PAVI-E scheme:

- $\mathbb{1}_{2 \in \mathcal{B}_1[t]}$ depends on whether the index 2 is chosen as part of sub-sampling the plate $\mathcal{P}_1$, and therefore only depends on the RV $I_{\mathcal{P}_1}$. In this case the associated probability is $2/3$;

- to evaluate $\log q_{1,2}(\theta_{1,2} | \theta_{2,0}; \phi_1, \mathbf{E}_{1,2})$, the value of $\mathbf{E}_{1,2}$ will result from the application of the encoder $f$ over the value of either $X_4$ or $X_5$. This choice depends on whether the index 0 or 1 is chosen as part of sub-sampling the plate $\mathcal{P}_0$. Therefore, the value of the term $\log q_{1,2}$ only depends on the RV $I_{\mathcal{P}_0}$.

In both PAVI-F and PAVI-E– the terms $\mathbb{1}_{n \in \mathcal{B}_i[t]}$ and $\log q_{i,n}(\theta_{i,n} | \pi(\theta_{i,n}); \phi_i, \mathbf{E}_{i,n})$ are therefore independent, meaning that the expectation of their product can be rewritten as the product of their

expectations:

$$
\begin{aligned}
\mathbb{E}_{I_{\mathcal{P}}}\left[\log q^{\mathfrak{r}}(\Theta^{\mathfrak{r}}[t])\right] &= \sum_{i=1}^{I} \frac{N_i}{N_i^{\mathfrak{r}}} \sum_{n=0}^{N_i} \mathbb{E}_{I_{\mathcal{P}}}\left[\mathbb{1}_{n \in \mathcal{B}_i[t]}\right] \mathbb{E}_{I_{\mathcal{P}}}\left[\log q_{i,n}(\theta_{i,n}|\pi(\theta_{i,n}))\right] \\
&= \sum_{i=1}^{I} \frac{N_i}{N_i^{\mathfrak{r}}} \sum_{n=0}^{N_i} \frac{N_i^{\mathfrak{r}}}{N_i} \mathbb{E}_{I_{\mathcal{P}}}\left[\log q_{i,n}(\theta_{i,n}|\pi(\theta_{i,n}))\right] \quad (\mathrm{A.11}) \\
&= \sum_{i=1}^{I} \sum_{n=0}^{N_i} \mathbb{E}_{I_{\mathcal{P}}}\left[\log q_{i,n}(\theta_{i,n}|\pi(\theta_{i,n}); \mathbf{E}_{i,n})\right]
\end{aligned}
$$

This equality can be further simplified in the PAVI-F case –proving its unbiasedness– but not in the PAVI-E case, as detailed in the sections below.

### A.2.2 UNBIASEDNESS OF THE PAVI-F SCHEME

In the PAVI-F scheme, detailed in section 2.4, the encodings $\mathbf{E}_{i,n}$ are constants with respect to the branching choice, therefore we have:

$$
\begin{aligned}
\mathbb{E}_{I_{\mathcal{P}}}\left[\log q^{\mathfrak{r}}(\Theta^{\mathfrak{r}}[t])\right] &= \sum_{i=1}^{I} \sum_{n=0}^{N_i} \mathbb{E}_{I_{\mathcal{P}}}\left[\log q_{i,n}(\theta_{i,n}|\pi(\theta_{i,n}); \mathbf{E}_{i,n})\right] \\
&= \sum_{i=1}^{I} \sum_{n=0}^{N_i} \log q_{i,n}(\theta_{i,n}|\pi(\theta_{i,n}); \mathbf{E}_{i,n}) \quad (\mathrm{A.12}) \\
&= \log q(\Theta)
\end{aligned}
$$

which proves eq. (A.8) and eq. (A.6). In the above example of $\theta_{1,2}$ in $\mathcal{M}$, in the PAVI-F scheme the expression $\mathbb{E}_{I_{\mathcal{P}}}\left[\log q_{i,n}(\theta_{i,n}|\pi(\theta_{i,n}); \phi_i, \mathbf{E}_{i,n})\right]$ can be evaluated into $\log q_{1,2}(\theta_{1,2}|\theta_{2,0}; \phi_1, \mathbf{E}_{1,2})$. This demonstrates that the PAVI-F scheme is unbiased: training over stochastically chosen subgraphs for $q^{\mathfrak{r}}$ is in expectation equal to training over the full graph of $q$.

### A.2.3 APPROXIMATIONS IN THE PAVI-E SCHEME

In the PAVI-E scheme, detailed in section 2.3.2, the encodings $\mathbf{E}_{i,n}$ are computed from the observed data $X$. Specifically, considering the ground RV $\theta_{i,n}$, we have $\mathbf{E}_{i,n} = f(\mathbf{X}_{i,n}^{\mathfrak{r}}[t])$ where $\mathbf{X}_{i,n}^{\mathfrak{r}}[t]$ corresponds to the observed data of a subset of $\theta_{i,n}$'s descendants. Depending on the chosen branching *downstream* of $\theta_{i,n}$, the value of $\mathbf{E}_{i,n}$ can therefore vary. This means we cannot further simplify eq. (A.11): the terms $\log q_{i,n}(\theta_{i,n}|\pi(\theta_{i,n}); \mathbf{E}_{i,n})$ are not constants with respect to the RVs $I_{\mathcal{P}}$. In the above example of $\theta_{1,2}$ in $\mathcal{M}$, in the PAVI-E scheme the expression $\mathbb{E}_{I_{\mathcal{P}}}\left[\log q_{i,n}(\theta_{i,n}|\pi(\theta_{i,n}); \phi_i, \mathbf{E}_{i,n})\right]$ can be evaluated into:

$$
\frac{1}{2}\left(\log q_{1,2}(\theta_{1,2}|\theta_{2,0}; \phi_1, f(\mathbf{X}_4)) + \log q_{1,2}(\theta_{1,2}|\theta_{2,0}; \phi_1, f(\mathbf{X}_5))\right)
$$

**How could the PAVI-E scheme be made unbiased?** Specifically, by making the value of $\mathbf{E}_{i,n}$ independent of the choice of downstream branching. A possibility would be to parameterize $\mathbf{E}_{i,n}$ as an average –an expectation– over all the possible sub-branchings downstream of $\theta_{i,n}$. Yet, in practical cases, the cardinalities of the reduced model are much inferior to the ones of the full model: $\mathrm{Card}^{\mathfrak{r}}(\mathcal{P}) \ll \mathrm{Card}(\mathcal{P})$. This means that numerous $\mathrm{Card}^{\mathfrak{r}}(\mathcal{P})$-sized subsets can be chosen inside the $\mathrm{Card}(\mathcal{P})$ possible descendants. In order to average over all those subset choices to compute $\mathbf{E}_{i,n}$, numerous encoding calculations would be required at each stochastic training step. For large-scale cases, we deemed this possibility impractical. Other possibilities could exist, all revolving around the problematic of aggregating collections of stochastic estimators into one general estimator –in an unbiased and efficient manner. To our knowledge, this is a complex and still open research question, whose advancement could much benefit our applications.

**Practical approximation for the PAVI-E scheme** In practice, we compute the encoding $\mathbf{E}_{i,n}$ based on the single downstream branching corresponding to the sampling of the RVs $I_{\mathcal{P}}$. Compared to

the previous paragraph, this amounts to estimating the expectation of $\mathbf{E}_{i,n}$ –over all downstream branchings– using a single one of those branchings. Note that, even if this encoding estimate was unbiased, $\log q_{i,n}$ would remain an highly non-linear function of $\mathbf{E}_{i,n}$. As a consequence, we need to rely on the approximation:

$$\mathbb{E}_{I_{\mathcal{P}}} \left[ \log q_{i,n}(\theta_{i,n}|\pi(\theta_{i,n}); \phi_i, f(\mathbf{X}_{i,n}^{\mathfrak{r}}[t])) \right] \simeq \log q_{i,n}(\theta_{i,n}|\pi(\theta_{i,n}); \phi_i, f(\mathbf{X}_{i,n})) \tag{A.13}$$

which can theoretically introduce some bias in our gradients. The approximation eq. (A.13) can be interpreted as follow: "the expectation of the density of $\theta_{i,n}$ when collecting summary statistics over a stochastic subset of $\theta_{i,n}$'s descendants is approximately equal to the density of $\theta_{i,n}$ when collecting summary statistics over the entirety of $\theta_{i,n}$'s descendants". Another interpretation is that the distribution associated to the summary of the full data can be approximated by an annealing of the distributions associated to summaries of subsets of this data. In practice, this approximation did not yield significantly worse performance for the PAVI-E scheme over the generative models we tested. At the same time, computing the encodings over a single branching allows to compute all the $\mathbf{E}_{i,n}$ encodings in a single lightweight pass over the data $\mathbf{X}^{\mathfrak{r}}[t]$. This simple solution therefore provided with a substantial increase in training speed with seldom noticeable bias. Yet, we do not bar the existence of pathological generative HBMs where this approximation would become coarse. Experimenters should bear in mind this possibility when using the PAVI-E scheme. In practice, using the PAVI-F scheme as a sanity check over synthetic, toy-dimension implementations of the considered generative models is a good way to validate the PAVI-E scheme –before moving onto the real problem instantiating the same generative model with a larger dimensionality.

## A.3 PAVI ALGORITHMS

More technical details can be found in the codebase provided with this supplemental material.

### A.3.1 ARCHITECTURE BUILD

---

**Algorithm 1:** PAVI architecture build

---

**Input:** Graph template $\mathcal{T}$, plate cardinalities $\{(\mathrm{Card}(\mathcal{P}_p), \mathrm{Card}^{\mathfrak{r}}(\mathcal{P}_p))\}_{p=0..P}$, encoding scheme

**Output:** $q$ distribution

**for** $i = 1..I$ **do**

    Construct conditional flow $\mathcal{F}_i$;

    Define conditional posterior distributions $q_{i,n}$ as the push-forward of the prior via $\mathcal{F}_i$, following eq. (2);

Combine the $q_{i,n}$ distributions following the cascading flows scheme, as in section 2.2.1 (Ambrogioni et al., 2021b) ;

**if** *PAVI-F encoding scheme* **then**

    Construct encoding arrays $\{\mathbf{E}_i = [\mathbf{E}_{i,n}]_{n=0..N_i}\}_{i=1..I}$ as in appendix A.1.2 ;

**else if** *PAVI-E encoding scheme* **then**

    Construct encoder $f$ as in appendix A.1.3 ;

---

### A.3.2 STOCHASTIC TRAINING

---

**Algorithm 2:** PAVI stochastic training

---

**Input:** Untrained architecture $q$, observed data $\mathbf{X}$, encoding scheme, number of steps $T$
**Output:** trained architecture $q$
**for** $t = 0..T$ **do**
    Sample plate indices to define the batches $\mathcal{B}_i[t]$, the latent $\Theta^{\mathfrak{r}}[t]$ and the observed $X^{\mathfrak{r}}[t]$ and
      $\mathbf{X}^{\mathfrak{r}}[t]$, following appendix A.1.1 ;
    Define reduced distribution $p^{\mathfrak{r}}$ ;
    **if** *PAVI-F encoding scheme* **then**
        Collect encodings $\mathbf{E}_{i,n}$ by slicing from the arrays $\mathbf{E}_i$ the elements corresponding to the
        batches $\mathcal{B}_i[t]$ ;
    **else if** *PAVI-E encoding scheme* **then**
        Compute encodings as $\mathbf{E} = f(\mathbf{X}^{\mathfrak{r}}[t])$;
    Feed obtained encodings into $q^{\mathfrak{r}}$ ;
    Compute reduced ELBO as in eq. (5), back-propagate its gradient ;
    Update conditional flow weights $\{\phi_i\}_{i=1..I}$;
    **if** *PAVI-F encoding scheme* **then**
        Update encodings $\{\mathbf{E}_{i,n}\}_{i=1..I, n \in \mathcal{B}_{i,t}}$;
    **else if** *PAVI-E encoding scheme* **then**
        Update encoder weights $\eta$;

---

### A.3.3 INFERENCE

---

**Algorithm 3:** PAVI inference

---

**Input:** trained architecture $q$, observed data $\mathbf{X}$, encoding scheme
**Output:** approximate posterior distribution
**if** *PAVI-F encoding scheme* **then**
    Collect full encoding arrays $\mathbf{E}_i$ ;
**else if** *PAVI-E encoding scheme* **then**
    Compute encodings as $\mathbf{E} = f(\mathbf{X})$ using set size generalization ;
Feed obtained encodings into $q$ ;

---

### A.4 INFERENCE GAPS

In terms of inference quality, the impact of our architecture can be formalized following the *gaps* terminology (Cremer et al., 2018). Consider a joint distribution $p(\Theta, X)$, and a value $\mathbf{X}$ for the RV template $X$. We pick a variational family $\mathcal{Q}$, and in this family look for the parametric distribution $q(\Theta; \phi)$ that best approximates $p(\Theta | X = \mathbf{X})$. Specifically, we want to minimize the Kulback-Leibler divergence (Blei et al., 2017) between our variational posterior and the true posterior, that Cremer et al. (2018) refer to as the *gap* $\mathcal{G}$:

$$\begin{aligned} \mathcal{G} &= \mathrm{KL}(q(\Theta; \phi) || p(\Theta | X)) \\ &= \log p(X) - \mathrm{ELBO}(q; \phi) \end{aligned} \tag{A.14}$$

We denote $q^*(\Theta; \phi^*)$ the optimal distribution inside $\mathcal{Q}$ that minimizes the KL divergence with the true posterior:

$$\begin{aligned} \mathcal{G}_{\mathrm{approx}}(\mathcal{Q}; \phi^*) &= \log p(X) - \mathrm{ELBO}(q^*; \phi^*) \\ &\geq 0 \end{aligned} \tag{A.15}$$

$$\mathcal{G}_{\mathrm{vanilla\ VI}} = \mathcal{G}_{\mathrm{approx}}$$

The *approximation gap* $\mathcal{G}_{\mathrm{approx}}$ depends on the expressivity of the variational family $\mathcal{Q}$, specifically whether $\mathcal{Q}$ contains distributions arbitrarily close to the posterior –in the KL sense.

**Note:** $\mathcal{G}_{\mathrm{approx}}$ is a property of the variational family $\mathcal{Q}$. $\mathcal{G}_{\mathrm{approx}}$ is an asymptotic bound for the KL divergence between any distribution $q \in \mathcal{Q}$ and the true posterior. This gap is therefore a form of bias, but is not to be mistaken with the stochasticity-induced bias studied in appendix A.2. The bias

in appendix A.2 relates to whether $q^*$ can be found by training stochastically over $\mathcal{Q}$, whereas $\mathcal{G}_{\text{approx}}$ relates to the the bias between $q^*$ and the true posterior.

Cremer et al. (2018) demonstrate that, in the case of sample amortized inference, when the weights $\phi$ no longer are free but the output of an encoder $f \in \mathcal{F}$, inference cannot be better than in the non-sample-amortized case, and a positive *amortization gap* is introduced:

$$\mathcal{G}_{\text{sa}}(\mathcal{Q}, \mathcal{F}; \eta^*) = \mathcal{G}_{\text{approx}}(\mathcal{Q}; f(\mathbf{X}, \eta^*)) - \mathcal{G}_{\text{approx}}(\mathcal{Q}; \phi^*)$$
$$\geq 0 \tag{A.16}$$
$$\mathcal{G}_{\text{sample amortized VI}} = \mathcal{G}_{\text{approx}} + \mathcal{G}_{\text{sa}}$$

Where we denote as $\eta^*$ the optimal weights for the encoder $f$ inside the function family $\mathcal{F}$. The gap terminology can be interpreted as follow: "theoretically, sample amortization cannot be beneficial in terms of KL divergence for the inference over a given sample $\mathbf{X}$."

Using the same gap terminology, we can define gaps implied by our PAVI architecture. Instead of picking the distribution $q$ inside the family $\mathcal{Q}$, consider picking $q$ from the *plate-amortized* family $\mathcal{Q}_{\text{pa}}$ corresponding to $\mathcal{Q}$. Distributions in $\mathcal{Q}_{\text{pa}}$ are distributions from $\mathcal{Q}$ with the additional constraints that some weights have to be equal. Consequently, $\mathcal{Q}_{\text{pa}}$ is a subset of $\mathcal{Q}$:

$$\mathcal{Q}_{\text{pa}} \subset \mathcal{Q} \tag{A.17}$$

As such, looking for the optimal distribution inside $\mathcal{Q}_{\text{pa}}$ instead of inside $\mathcal{Q}$ cannot result in better performance, leading to a *plate amortization gap*:

$$\mathcal{G}_{\text{pa}}(\mathcal{Q}, \mathcal{Q}_{\text{pa}}; \psi^*, \phi^*) = \mathcal{G}_{\text{approx}}(\mathcal{Q}_{\text{pa}}; \psi^*) - \mathcal{G}_{\text{approx}}(\mathcal{Q}; \phi^*)$$
$$\geq 0 \tag{A.18}$$
$$\mathcal{G}_{\text{PAVI-F}} = \mathcal{G}_{\text{approx}} + \mathcal{G}_{\text{pa}}$$

Where we denote as $\psi^*$ the optimal weights for a variational distribution $q$ inside $\mathcal{Q}_{\text{pa}}$ –in the KL sense. The equation A.18 is valid for the PAVI-F scheme –see section 2.4. We can interpret it as follow: "theoretically, plate amortization cannot be beneficial in terms of KL divergence for the inference over a given sample $\mathbf{X}$".

Now consider that encodings are no longer free parameters but the output of an encoder $f$. Similar to the case presented in eq. (A.16), using an encoder cannot result in better performance, leading to an *encoder gap*:

$$\mathcal{G}_{\text{encoder}}(\mathcal{Q}_{\text{pa}}, \mathcal{F}; \psi^*, \eta^*) = \mathcal{G}_{\text{approx}}(\mathcal{Q}_{\text{pa}}; f(\mathbf{X}, \eta^*)) - \mathcal{G}_{\text{approx}}(\mathcal{Q}_{\text{pa}}; \psi^*)$$
$$\geq 0 \tag{A.19}$$
$$\mathcal{G}_{\text{PAVI-E}} = \mathcal{G}_{\text{approx}} + \mathcal{G}_{\text{pa}} + \mathcal{G}_{\text{encoder}}$$

The equation eq. (A.19) is valid for the PAVI-E scheme –see section 2.4.

The most complex case is the PAVI-E(sa) scheme, where we combine both plate and sample amortization. Our argument cannot account for the resulting $\mathcal{G}_{\text{PAVI-E(sa)}}$ gap: both the PAVI-E and PAVI-E(sa) schemes rely upon the same encoder $f$. In the PAVI-E scheme, $f$ is overfit over a dataset composed of the slices of a given data sample $\mathbf{X}$. In the PAVI-E(sa) scheme, the encoder is trained over the whole distribution of the samples of the reduced model $\mathcal{M}^{\mathfrak{r}}$. Intuitively, it is likely that the performance of PAVI-E(sa) will always be dominated by the performance of PAVI-E, but –as far as we understand it– the gap terminology cannot account for this discrepancy.

Comparing previous equations, we therefore have:

$$\mathcal{G}_{\text{vanilla VI}} \leq \mathcal{G}_{\text{PAVI-F}} \leq \mathcal{G}_{\text{PAVI-E}} \tag{A.20}$$

Note that those are *theoretical* results, that do not necessarily pertain to optimization in practice. In particular, in section 3.1&3.3, this theoretical performance loss is not observed empirically over the studied examples. On the contrary, in practice our results can actually be better than non-amortized baselines, as is the case for the PAVI-F scheme in fig. 4 or experiments B.3. We interpret this as a result of a simplified optimization problem due to plate amortization –with fewer parameters to optimize for, and mini-batching effects across different ground RVs. A better framework to explain those discrepancies could be the one from Bottou & Bousquet (2007): performance in practice is not only the reflection of an *approximation error*, but also of an *optimization error*. A less expressive architecture –using plate amortization– may in practice yield better performance. Furthermore, for the experimenter, the theoretical gaps $\mathcal{G}_{\text{pa}}, \mathcal{G}_{\text{encoder}}$ are likely to be well "compensated for" by the lighter parameterization and faster convergence entitled by plate amortization.

# B SUPPLEMENTAL RESULTS

## B.1 GRE RESULTS SANITY CHECK

As exposed in the introduction of section 3, in this work we focused on the usage of the ELBO as an inference performance metric (Blei et al., 2017):

$$\text{ELBO}(q) = \log p(X) - \text{KL}(q(\Theta)||p(\Theta|X)) \tag{B.21}$$

Given that the likelihood term $\log p(X)$ does not depend on the variational family $q$, differences in ELBOs directly transfer in differences in KL divergence, and provide with a straightforward metric to compare different variational posteriors. Nonetheless, the ELBO doesn't provide with an absolute metric of quality. As a sanity check, we want to assert the quality of the results presented in section 3.3 –that are transferable to section 3.1&3.2, based on the same model. In fig. B.1 we plot the posterior samples of various methods against analytical ground truths, using the $\text{Card}(\mathcal{P}_1) = 20$ case. All the method's results are aligned with the analytical ground truth, with differences in ELBO translating meaningful qualitative differences in terms of inference quality.

## B.2 EFFECT OF THE REDUCED MODEL CARDINALITIES ON THE TRAINING EFFICIENCY

In fig. B.2 we show the impact of the augmentation of $\text{Card}^{\mathfrak{r}}(\mathcal{P}_1)$ on the efficiency of the variational posterior's optimization.

In practice, we noticed that the most efficient choice in the case of the PAVI-F scheme was to maximize the cardinalities of the reduced model given the memory constraints of the GPU. Indeed, training over larger cardinalities does make each optimization step slightly slower but also make the ELBO gradient estimates less noisy and allows to train more encoding vectors $\mathbf{E}_{i,n}$ at a given optimization step.

In the case of the PAVI-E scheme, the training speed is constant with respect to $\text{Card}^{\mathfrak{r}}(\mathcal{P}_1)$. This is due to the compute of the encodings being vectorized across plates in our deep-set encoder – see appendix A.1.3 (Lee et al., 2019). We observe a slight if barely noticeable reduction of the inference bias when augmenting $\text{Card}^{\mathfrak{r}}(\mathcal{P}_1)$. Similar to the PAVI-F scheme, $\text{Card}^{\mathfrak{r}}(\mathcal{P}_1)$ should be maximized with respect to the GPU memory constraint. The intuition behind this choice is that the generalization of the learning of summary statistics across sets of data points is easier the closer the reduced set size is to the full set size. This constant training speed allows for a controlled memory footprint of the stochastic training: contrary to the PAVI-F scheme, $\text{Card}^{\mathfrak{r}}(\mathcal{P}_1)$ could be maintained at a value manageable by the GPU without reducing the training speed. This example also displays a pathological choice for $\text{Card}^{\mathfrak{r}}(\mathcal{P}_1)$: the encoder fails to learn to compute the correct summary statistics over sets of size 1. However trivial, this example underlines that $\text{Card}^{\mathfrak{r}}(\mathcal{P}_1)$ should be chosen at a value logical with respect to the posterior's sufficient statistics. When generalizing to moments of higher order –such as the variance– there is therefore a theoretical lower bound to consider when fixating the value of $\text{Card}^{\mathfrak{r}}(\mathcal{P}_1)$. Our intuition is that the more complex to estimate the statistic is, the larger $\text{Card}^{\mathfrak{r}}(\mathcal{P}_1)$ should be.

Note that in practice non-stochastic VI is intractable for large-scale models, because of its memory requirement. In large scale experiments such as the one presented in fig. 5, stochastic training is necessary. Our claim is to be faster than non-plate-amortized stochastic VI in those large-scale contexts –but not necessarily to be faster then non-stochastic VI in the small-scale regime of this experiment. For PAVI-F we nonetheless obtain similar training speed compared to non-stochastic VI with $\text{Card}^{\mathfrak{r}}(\mathcal{P}_1)$ as small as 8. For PAVI-E, the stochastic training is as fast as the non-stochastic one.

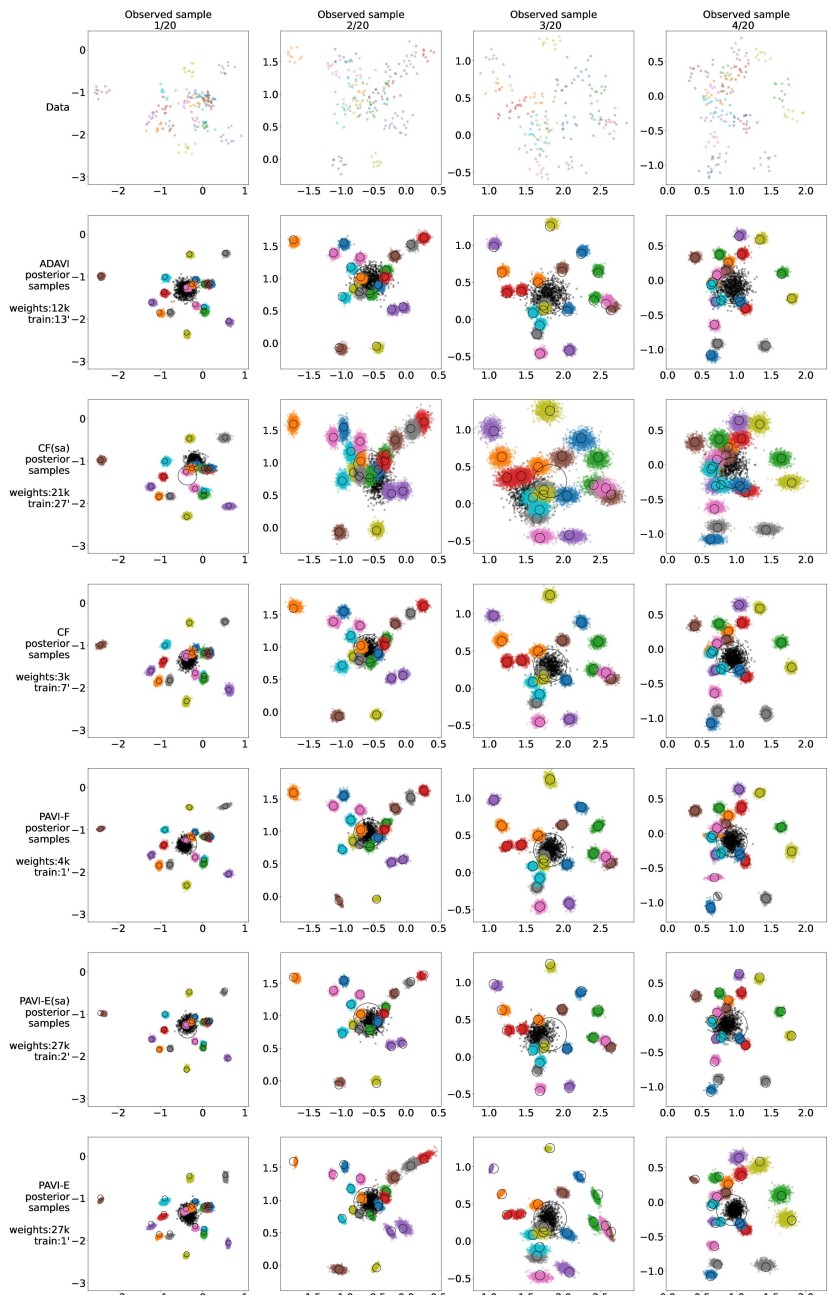

Figure B.1: **GRE Sanity check** Inference methods present qualitatively correct results, making ELBO comparisons relevant in our experiments. *On the topmost line*, we represent 4 different **X** samples for the GRE model described in eq. (6) with $\mathrm{Card}(\mathcal{P}_1) = 20$. Each set of colored points represent the $\mathbf{X}_{n_1,\bullet}$ points belonging to one of the 20 groups. *Bottom lines* represent the posterior samples for the methods used in section 3.3. Colored points are sampled from the posterior of the groups means $\theta_1$, whereas black points are samples from the population mean $\theta_2$. We represent as black circles an analytical ground truth, centered on the correct posterior mean, and with a radius equal to 2 times the analytical posterior's standard deviation. **Correct posterior samples should be centered on the same point as the corresponding black circle, and 95% of the points should fall within the black circle**. PAVI is represented on the 3 last lines. Some minor bias can be observed for the PAVI-E scheme, but this approximation error is marginal compared to the optimization error that can be observed for unbiased methods, such as CF(sa) (Bottou & Bousquet, 2007). We can observed a superior quality for the PAVI-F scheme, rivaling ADAVI and CF's performance with orders of magnitude less parameters and training time, as visible in fig. 4.

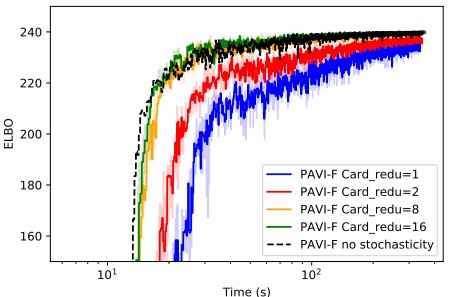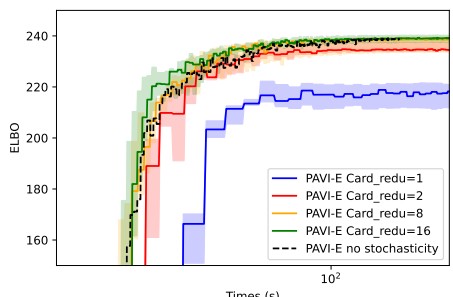

Figure B.2: **Both panels: Effect of** $\mathrm{Card}^{\mathfrak{r}}(\mathcal{P}_1)$ **on the training efficiency** Experiment performed on the GRE model (see eq. (6)). We increase the cardinality of the reduced model $\mathrm{Card}^{\mathfrak{r}}(\mathcal{P}_1)$ from 1 to 20 while keeping $\mathrm{Card}(\mathcal{P}_1) = 20$ fixed. At $\mathrm{Card}^{\mathfrak{r}}(\mathcal{P}_1) = 20$, there is no stochasticity in the training, meaning we train directly on $\mathcal{M}$. This experiment is interesting to evaluate the bias introduced in the stochastic training, as detailed in appendix A.2. **Left panel: PAVI-F scheme** As $\mathrm{Card}^{\mathfrak{r}}(\mathcal{P}_1)$ augments, the training gets faster and less noisy, likely due to less stochasticity in the gradient estimates and more encodings vectors $\mathbf{E}_{i,n}$ being trained at once. This increase in speed quickly caps, and the speed is approximately the same between $\mathrm{Card}^{\mathfrak{r}}(\mathcal{P}_1) = 8$ and $\mathrm{Card}^{\mathfrak{r}}(\mathcal{P}_1) = 16$. This experiment also illustrates the unbiasedness of the stochastic training: at $\mathrm{Card}^{\mathfrak{r}}(\mathcal{P}_1) = 20$ there is no stochasticity in the training, and the asymptotic performance is the same as for the stochastic training (see appendix A.2). **Right panel: PAVI-E scheme** The case of $\mathrm{Card}^{\mathfrak{r}}(\mathcal{P}_1) = 1$ is pathological: the encoder "learns" to collect summary statistics across a set of 1 element, and –not surprisingly– the learnt function doesn't generalize well on sets of size 20. In all of the other cases, the ELBO converges approximately to the same values as with $\mathrm{Card}^{\mathfrak{r}}(\mathcal{P}_1) = 20$, that is to say when there is no stochasticity in the training (black curve). We furthermore observe a slight reduction of the inference bias as $\mathrm{Card}^{\mathfrak{r}}(\mathcal{P}_1)$ augments. This illustrates how the theoretical bias of the PAVI-E scheme identified in appendix A.2 does not translate in significantly worse empirical results –albeit in the pathological case of $\mathrm{Card}^{\mathfrak{r}}(\mathcal{P}_1) = 1$. Interestingly, the training speed is approximately constant across all values of $\mathrm{Card}^{\mathfrak{r}}(\mathcal{P}_1)$. This feature is essential in the PAVI-E scheme: we can train over a reduced version of our model –with a light memory footprint– and apply the obtained architecture to the full model. Note that this feature wouldn't necessarily be present when training on CPU: computing summary statistics over larger sets would make the training *slower* when $\mathrm{Card}^{\mathfrak{r}}(\mathcal{P}_1)$ augments.

| Architecture | ELBO ($10^1$) | Optimization time (s) |
|---|---|---|
| CF | - 16.2 ($\pm$ 2.9) | 3,100 |
| ADAVI | - 20.0 ($\pm$ 2.3) | 3,000 |
| UIVI | - 26.3 ($\pm$ 1.2) | 500 |
| PAVI-E (ours) | -15.2 ($\pm$2.8) | 470 |
| PAVI-F (ours) | **- 11.1 ($\pm$1.1)** | **300** |

Table 1: **Analytical performance over a Gaussian mixture HBM** PAVI shows superior performance. PAVI converges in a fraction of CF's optimization time –the second best performing architecture.

### B.3 ADDITIONAL COMPARATIVE EXPERIMENTS

#### B.3.1 GAUSSIAN MIXTURE MODEL

In this experiment we test out various baselines over a challenging model: a Gaussian mixture with random effects.

$$D, \quad \text{Card}(\mathcal{P}_1), \quad \text{Card}(\mathcal{P}_0) = 2, \quad 20, \quad 10$$

$$\begin{aligned}
&{}^{\forall n_1 = 1..\,\text{Card}(\mathcal{P}_1)}_{\forall n_0 = 1..\,\text{Card}(\mathcal{P}_0)} \quad X_{n_1,n_0} | \theta_{1,n_1}, \pi_{n_1} \sim \text{Mixture}\big(\big[\mathcal{N}(\theta^1_{1,n_1}, \sigma_x^2), \ldots, \mathcal{N}(\theta^L_{1,n_1}, \sigma_x^2)\big], \pi_{n_1}\big) \\
&\forall n_1 = 1..\,\text{Card}(\mathcal{P}_1) \quad \pi_{n_1} \sim \text{Dirichlet}(1 \times \vec{1}_L) \\
&{}^{\forall n_1 = 1..\,\text{Card}(\mathcal{P}_1)}_{\forall l = 1..L} \quad \theta^l_{1,n_1} | \theta^l_{2,0} \sim \mathcal{N}(\theta^l_{2,0}, \sigma_1^2) \\
&\forall l = 1..L \quad \theta^l_{2,0} \sim \mathcal{N}(\vec{0}_D, \sigma_2^2) \ ,
\end{aligned}$$
(B.22)

where $\text{Mixture}([\mathcal{D}_1, \ldots, \mathcal{D}_L], \pi)$ denotes a mixture between the distributions $[\mathcal{D}_1, \ldots, \mathcal{D}_L]$ with mixture weights $\pi$. Results are visible in table 1, where PAVI displays the best asymptotic ELBO, as well as the shortest optimization time. On that note, we underline that stochastic training over a mixture distribution is challenging, as –due to the sub-sampling of points– only a fraction of the mixture components could be expressed at a given step, requiring the architecture to dynamically cluster the data points across time.

#### B.3.2 HIERARCHICAL VARIANCE MODEL

In this experiment we test out our architectures over a non-canonical model, in which parent RVs play the role of variance for the distribution of their children. Our goal is in particular to evaluate a potential empirical bias for the PAVI-E scheme –as studied in appendix A.2. We will refer the following HBM as the Hierarchical Variance model:

$$D, \quad \text{Card}(\mathcal{P}_1), \quad \text{Card}(\mathcal{P}_0) = 2, \quad 15, \quad 15$$

$$\begin{aligned}
&{}^{\forall n_1 = 1..\,\text{Card}(\mathcal{P}_1)}_{\forall n_0 = 1..\,\text{Card}(\mathcal{P}_0)} \quad \log X_{(n_1, n_0)} | \theta_{1,n_1} \sim \mathcal{N}(0, \theta_{1,n_1}) \\
&\forall n_1 = 1..\,\text{Card}(\mathcal{P}_1) \quad \log \theta_{1,(n_1)} | \theta_{2,0} \sim \mathcal{N}(0, \theta_{0,0}) \\
&\log \theta_{2,0} \sim \mathcal{N}(\vec{0}_D, 1)
\end{aligned}$$
(B.23)

In this model, the encoder –used in the PAVI-E scheme– has to collect non-trivial summary statistics: empirical variances across variable subsets of the observed data. This is the context in which we would expect to observe the most bias due to the approximation in eq. (A.13). The performance of PAVI-E remains nonetheless competitive. This illustrates how PAVI-E's theoretical bias –introduced in the stochastic training– does not result in significantly worse inference compared to state-of-the-art architectures. PAVI also displays the best ELBO, in conjunction with UIVI, but in a 5 times shorter optimization time.

#### B.3.3 SMALL DIMENSION VERSION OF OUR NEUROIMAGING MODEL

In this experiment we validate the performance of our architecture on synthetic data generated using a small dimension version of the model presented in eq. (B.24). To allow the use of the comparative

| Architecture | ELBO ($10^2$) | Optimization time (s) |
|---|---|---|
| CF | - 11.2 ($\pm$ 2.9) | 500 |
| UIVI | -6.7 ($\pm$ 0.8) | 100 |
| PAVI-E (ours) | -6.7 ($\pm$1.0)[1] | 90 |
| PAVI-F (ours) | **- 6.7 ($\pm$0.8)** | **20** |

Table 2: **Analytical performance over a Hierarchical Variance HBM** In this setting with hard-to-estimate summary statistics, PAVI-E doesn't show any empirical bias. [1]: PAVI-E suffers from numerical instability on this example, with runs degenerating into NaN results.

| Architecture | ELBO ($10^4$) | Optimization time (s) |
|---|---|---|
| CF | - 139.7 ($\pm$ 21) | 3,200 |
| ADAVI | — | — |
| UIVI | — | — |
| PAVI-F (ours) | **- 8.2 ($\pm$ 0.1)** | **1,600** |

Table 3: **Analytical performance over our Neuroimaging HBM** This HBM is not pyramidal and consequently cannot be processed by the ADAVI architecture. Due to the large parameter space (approximately 2k parameters) and the complex density involved, despite intensive efforts we did not manage to obtain a numerically stable UIVI optimization. PAVI shows superior compared to CF, in half its optimization time.

baselines, we fixate $S, S^{\mathfrak{r}} = 5, 3, T, T^{\mathfrak{r}} = 6, 3, N, N^{\mathfrak{r}} = 12, 3, D = 4, L = 7$. Results are visible in table 3: PAVI can deal with HBMs unavailable to the ADAVI architecture (Rouillard & Wassermann, 2022), and so so with a performance superior to the CF baseline (Ambrogioni et al., 2021b). Despite intensive efforts we did not manage to obtain a numerically stable UIVI optimization. This is likely due to the complexity of this inference problem: a large dimensionality (2k parameters) combined with complex dependencies between RVs in the posterior. In this context, automatic structured VI baselines such as CF or PAVI exploit the parametric form of the prior $p$ to help with inference. PAVI's plate amortization likely facilitates the inference, resulting in a reduced *optimization error* (Bottou & Bousquet, 2007).

## B.4 EXPERIMENTAL DETAILS - ANALYTICAL EXAMPLES

All experiments were performed in Python, using the *Tensorflow Probability* library (Dillon et al., 2017). Throughout this section we refer to *Masked Autoregressive Flows* (Papamakarios et al., 2018) as *MAF*. All experiments are performed using the Adam optimizer (Kingma & Ba, 2015). At training, the ELBO was estimated using a Monte Carlo procedure with 8 samples. All architectures were evaluated over a fixed set of 20 samples **X**, with 5 seeds per sample. Non-sample-amortized architectures were trained and evaluated on each of those points. Sample amortized architectures were trained over a dataset of $20,000$ samples separate from the 20 validation samples, then evaluated over the 20 validation samples.

### B.4.1 PLATE AMORTIZATION AND CONVERGENCE SPEED (3.1)

All 3 architectures (baseline, PAVI-F, PAVI-E) used:

- for the flows $\mathcal{F}_i$, a MAF with $[32, 32]$ hidden units;

- as encoding size, 128

For the encoder $f$ in the PAVI-E scheme, we used a multi-head architecture with 4 heads of 32 units each, 2 ISAB blocks with 64 inducing points.

### B.4.2    IMPACT OF ENCODING SIZE (3.2)

All architectures used:

- for the flows $\mathcal{F}_i$, a MAF with $[32, 32]$ hidden units, after an affine block with triangular scaling matrix.
- as encoding size, a value varying from 2 to 16

### B.4.3    SCALING WITH PLATE CARDINALITIES (3.3)

**ADAVI** (Rouillard & Wassermann, 2022) we used:

- for the flows $\mathcal{F}_i$, a MAF with $[32, 32]$ hidden units, after an affine block with triangular scaling matrix.
- for the encoder, an encoding size of 8 with a multi-head architecture with 2 heads of 4 units each, 2 ISAB blocks with 32 inducing points.

**Cascading Flows** (Ambrogioni et al., 2021b) we used:

- a mean-field distribution over the auxiliary variables $r$
- as auxiliary size, a fixed value of 8
- as flows, *Highway Flows* as designed by the Cascading Flows authors

**PAVI-F** we used:

- for the flows $\mathcal{F}_i$, a MAF with $[32, 32]$ hidden units, after an affine block with triangular scaling matrix.
- an encoding size of 8

**PAVI-E** we used:

- for the flows $\mathcal{F}_i$, a MAF with $[32, 32]$ hidden units, after an affine block with triangular scaling matrix.
- for the encoder, an encoding size of 16 with a multi-head architecture with 2 heads of 8 units each, 2 ISAB blocks with 64 inducing points.

**UIVI** we used as $\mathrm{Card}(\mathcal{P}_1) = 2 \rightarrow 20 \rightarrow 200$:

- as base distribution, a standard Gaussian with dimensionality $6 \rightarrow 42 \rightarrow 402$
- as transform $h$, an affine transform with diagonal scale
- as embedding distribution, a standard Gaussian with dimensionality $3 \rightarrow 6 \rightarrow 9$
- as transform weights regressor, a MLP with hidden units $[32, 32] \rightarrow [64, 64] \rightarrow [128, 128]$
- to sample uncorrelated samples $\epsilon$, a HMC run with 5 burn-in steps and 5 samples
- an Adam optimizer with exponential learning rate decay, starting at $1e-2$, $\times 0.9$ every 300 steps

### B.4.4    GAUSSIAN MIXTURE (B.3.1)

**ADAVI** (Rouillard & Wassermann, 2022) we used:

- for the flows $\mathcal{F}_i$, a MAF with $[32]$ hidden units, after an affine block with diagonal scaling matrix.
- for the encoder, an encoding size of 16 with a multi-head architecture with 2 heads of 8 units each, 2 ISAB blocks with 8 inducing points.

**Cascading Flows** (Ambrogioni et al., 2021b) we used:

- a mean-field distribution over the auxiliary variables $r$
- as auxiliary size, a fixed value of 16
- as flows, *Highway Flows* as designed by the Cascading Flows authors

**PAVI-F** we used:

- for the flows $\mathcal{F}_i$, a MAF with $[32]$ hidden units, after an affine block with diagonal scaling matrix.
- an encoding size of 16
- $\text{Card}^r(\mathcal{P}_1) = 5$

**PAVI-E** we used:

- for the flows $\mathcal{F}_i$, a MAF with $[128, 128]$ hidden units, after an affine block with traingular scaling matrix.
- for the encoder, an encoding size of 128 with a multi-head architecture with 4 heads of 32 units each, 2 ISAB blocks with 128 inducing points.
- $\text{Card}^r(\mathcal{P}_1) = 5$

**UIVI** we used:

- as base distribution, a standard Gaussian with dimensionality 82
- as transform $h$, an affine transform with diagonal scale
- as embedding distribution, a standard Gaussian with dimensionality 6
- as transform weights regressor, a MLP with hidden units $[64, 64]$
- to sample uncorrelated samples $\epsilon$, a HMC run with 5 burn-in steps and 5 samples
- an Adam optimizer with exponential learning rate decay, starting at $1e-2$, $\times 0.9$ every 300 steps

### B.4.5 HIERARCHICAL VARIANCES (B.3.2)

**Cascading Flows** (Ambrogioni et al., 2021b) we used:

- a mean-field distribution over the auxiliary variables $r$
- as auxiliary size, a fixed value of 32
- as flows, *Highway Flows* as designed by the Cascading Flows authors

**PAVI-F** we used:

- for the flows $\mathcal{F}_i$, a MAF with $[32, 32]$ hidden units, after an affine block with diagonal scaling matrix.
- an encoding size of 32
- $\text{Card}^r(\mathcal{P}_1) = 3$
- $\text{Card}^r(\mathcal{P}_0) = 3$

**PAVI-E** we used:

- for the flows $\mathcal{F}_i$, a MAF with $[128, 128]$ hidden units, after an affine block with diagonal scaling matrix.
- for the encoder, an encoding size of 128 with a multi-head architecture with 4 heads of 32 units each, 2 ISAB blocks with 128 inducing points.
- $\text{Card}^r(\mathcal{P}_1) = 3$
- $\text{Card}^r(\mathcal{P}_0) = 3$

**UIVI** we used:

- as base distribution, a standard Gaussian with dimensionality 32
- as transform $h$, an affine transform with diagonal scale
- as embedding distribution, a standard Gaussian with dimensionality 3
- as transform weights regressor, a MLP with hidden units $[64, 64]$
- to sample uncorrelated samples $\epsilon$, a HMC run with 5 burn-in steps and 5 samples
- an Adam optimizer with exponential learning rate decay, starting at $1e - 2$, $\times 0.9$ every 300 steps

### B.4.6 SMALL NEUROIMAGING EXAMPLE (B.3.3)

**Cascading Flows** (Ambrogioni et al., 2021b) we used:

- a mean-field distribution over the auxiliary variables $r$
- as auxiliary size, a fixed value of 32
- as flows, *Highway Flows* as designed by the Cascading Flows authors

**PAVI-F** we used:

- for the flows $\mathcal{F}_i$, a MAF with $[32, 32]$ hidden units, after an affine block with diagonal scaling matrix.
- an combination of encoding sizes of 32 and 8

**UIVI** we used:

- as base distribution, a standard Gaussian with dimensionality $1, 802$
- as transform $h$, an affine transform with diagonal scale
- as embedding distribution, a standard Gaussian with dimensionality 16
- as transform weights regressor, a MLP with hidden units $[128, 256, 512, 1024]$
- to sample uncorrelated samples $\epsilon$, a HMC run with 5 burn-in steps and 5 samples
- an Adam optimizer with exponential learning rate decay, starting at $1e - 3$, $\times 0.9$ every 300 steps

Optimization systematically degenerated into NaN results after around 200 optimization steps.

### B.5 DETAILS ABOUT OUR NEUROIMAGING EXPERIMENT (3.4)

### B.5.1 DATA DESCRIPTION

In this experiment we use data from the *Human Connectome Project (HCP)* dataset (Van Essen et al., 2012). We randomly select a cohort of $S = 1, 000$ subjects from this dataset, each subject being associated with $T = 2$ resting state fMRI sessions (Smith et al., 2013). We minimally pre-process the signal using the nilearn python library (Abraham et al., 2014):

1. removing high variance confounds
2. detrending the data
3. band-filtering the data (0.01 to 0.1 Hz), with a repetition time of 0.74 seconds
4. spatially smoothing the data with a 4mm Full-Width at Half Maximum

For every subject, we extract the surface Blood Oxygenation Level Dependent (BOLD) signal of $N = 314$ vertices corresponding to an average Broca's area (Heim et al., 2009). We compare this signal with the extracted signal of $D = 64$ DiFuMo components: a dictionary of brain spatial maps allowing for an effective fMRI dimensionality reduction (Dadi et al., 2020). Specifically,

we compute the one-to-one Pearson's correlation coefficient of every vertex with every DiFuMo component. The resulting connectome, with $S$ subjects, $T$ sessions, $N$ vertices and a connectivity signal with $D$ dimensions, is of shape $(S \times T \times N \times D)$. We project this data –correlation coefficients lying in $]-1; 1[$– in an unbounded space using an inverse sigmoid function.

### B.5.2 MODEL DESCRIPTION

We use a model inspired from the work of Kong et al. (2019). We hypothesize that every vertex in Broca's area belongs to either one of $L = 2$ functional networks. This functional bi-partition would reflect the anatomical partition between *pars opercularis* and *pars triangularis* (Heim et al., 2009; Zhang et al., 2020).

Each network is a pattern of connectivity with the brain cortex, represented as a the correlation of the BOLD signal with the signal from the $D = 64$ DiFuMo components. We define $L = 2$ such functional networks at the population level, that correspond to some "average" across the cohort of subjects. Every subject has an individual connectivity, and therefore individual $L = 2$ networks, that are considered as a Gaussian perturbation of the population networks, with variance $\epsilon$. The connectivity of a given subject also evolves through time, giving rise to session-specific networks, that are a Gaussian perturbation of the subject networks with variance $\sigma$. Finally, every vertex in Broca's area has its individual connectivity, and is a perturbation of one network's connectivity or the other's. We model this last step as a Gaussian mixture distribution with variance $\kappa$. We explicitly model the label label of a given vertex, and we consider this label constant across sessions.

The resulting model can be described as:

$$S, T, N, D, L = 1000, 2, 314, 64, 2$$
$$s^-, s^+ = -6, 0$$
$$\forall l=1..L: \quad \mu_l \sim \mathrm{Uniform}(-4 \times \vec{1}_D, 4 \times \vec{1}_D)$$
$$\forall l=1..L: \quad \log \epsilon_l \sim \mathrm{Uniform}(s^- \times \vec{1}_D, s^+ \times \vec{1}_D)$$
$$\substack{\forall l=1..L \\ \forall s=1..S}: \quad \mu_{l,s}|\mu_l, \epsilon_l \sim \mathcal{N}(\mu_l, \epsilon_l)$$
$$\forall l=1..L: \quad \log \sigma_l \sim \mathrm{Uniform}(s^- \times \vec{1}_D, s^+ \times \vec{1}_D) \quad \text{(B.24)}$$
$$\substack{\forall l=1..L \\ \forall s=1..S \\ \forall t=1..T}: \quad \mu_{l,s,t}|\mu_{l,s}, \sigma_l \sim \mathcal{N}(\mu_{l,s}, \sigma_l)$$
$$\forall l=1..L: \quad \log \kappa_l \sim \mathrm{Uniform}(s^- \times \vec{1}_D, s^+ \times \vec{1}_D)$$
$$\substack{\forall s=1..S \\ \forall n=1..N}: \quad \mathrm{probs}_{s,n} \sim \mathrm{Dirichlet}(1 \times \vec{1}_L)$$
$$\substack{\forall s=1..S \\ \forall n=1..N}: \quad \mathrm{label}_{s,n} \,|\, \mathrm{probs}_{s,n} \sim \mathrm{Categorical}(\mathrm{probs}_{s,n})$$
$$\substack{\forall s=1..S \\ \forall t=1..T \\ \forall n=1..N}: \quad X_{s,t,n}|[\mu_{l,s,t}]_{l=1..L}, [\kappa_l]_{l=1..L}, \mathrm{label}_{s,n} \sim \mathcal{N}(\mu_{\mathrm{label}_{s,n},s,t}, \kappa_{\mathrm{label}_{s,n}})$$

The model contains 4 plates: the *network* plate of full cardinality $L$ (that we did not exploit in our implementation), the *subject* plate of full cardinality $S$, the *session* plate of full cardinality $T$ and the *vertex* plate of full cardinality $N$.

Our goal is to recover the posterior distribution of the networks $\mu$ –represented as networks in fig. 5– and the labels label –represented as the parcellation in fig. 5– given the observed connectome described in appendix B.5.1.

### B.5.3 PAVI IMPLEMENTATION

We used in this experiment the PAVI-F scheme, using:

- for the RVs $\mu_l, \mu_{l,s}, \mu_{l,s,t}$:
  - for the flows $\mathcal{F}_i$, a MAF with $[128, 128]$ hidden units, following an affine block with diagonal scale
  - for the encoding size: 128
- for the RVs $\epsilon_l, \sigma_l, \kappa_l, \mathrm{probs}_{s,n}, \mathrm{labels}_{s,n}$:

- for the flows $\mathcal{F}_i$, a MAF with $[8, 8]$ hidden units, following an affine block with diagonal scale
    - for the encoding size: $8$
- for the reduced model, we used $S^{\mathfrak{r}} = 30$, $T^{\mathfrak{r}} = 1$ and $N^{\mathfrak{r}} = 32$.

To allow for the optimization over the discrete $\mathrm{label}_{s,n}$ RV, we used the Gumbell-Softmax trick, using a fixed temperature of 1.0 (Jang et al., 2017; Maddison et al., 2016).

## C  SUPPLEMENTAL DISCUSSION

### C.1  PLATE AMORTIZATION AS A GENERALIZATION OF SAMPLE AMORTIZATION

In section 2.2.2 we introduced plate amortization as the application of the generic concept of amortization to the granularity of plates. There is actually an even stronger connection between sample amortization and plate amortization.

A HBM $p$ models the distribution of a given observed RV $X$ –jointly with the parameters $\Theta$. Different samples $\mathbf{X}_0, \mathbf{X}_1, ...$ of the model $p$ are i.i.d. draws from the distribution $p(X)$. $p$ can thus be considered as the model for "one sample". Consider, instead of $p$, a "macro" model for the whole *population* of samples one could draw from $p$. The observed RV of that macro model would be the infinite collection of samples drawn from the same distribution $p(X)$. In that light, the i.i.d. sampling of different $X$ values from $p$ could be interpreted as a plate of the macro model. Thus, we could consider sample amortization as a instance of plate amortization for the "sample plate". Or equivalently: plate amortization can be seen as the generalization of amortization beyond the particular case of sample amortization.

### C.2  ALTERNATE FORMALISM FOR SVI – PAVI-E(SA) SCHEME

In this work, we propose a different formalism for SVI, based around the concept of full HBM $\mathcal{M}^{\mathrm{full}}$ versus reduced HBM $\mathcal{M}^{\mathrm{redu}}$ sharing the same template $\mathcal{T}$. This formalism is helpful to set up GPU-accelerated stochastic VI (Dillon et al., 2017), as it entitles a fixed computation graph -with the cardinality of the reduced model $\mathcal{M}^{\mathrm{redu}}$- in which encodings are "plugged in" -either sliced from larger encoding arrays or as the output of an encoder applied to a data slice, see section 2.2.1&2.3.2. Particularly, our formalism doesn't entitle a control flow over models and distributions, which can be hurtful in the context of *compiled* computation graphs such as in *Tensorflow* (Abadi et al., 2015).

The reduced model formalism is also meaningful in the PAVI-E(sa), where we train and amortized variational posterior over $\mathcal{M}^{\mathrm{redu}}$ and obtain "for free" a variational posterior for the full model $\mathcal{M}^{\mathrm{full}}$ –see section 2.3.2. In this context, our scheme is no longer a different take on hierarchical, batched SVI: the cardinality of the full model is truly independent from the cardinality of the training, and is only simulated as a scaling factor in the stochastic training –see section 2.3.1. We have the intuition that fruitful research directions could stem from this concept.

### C.3  BENEFITING FROM STRUCTURE IN INFERENCE

Our contributions can be abstracted through the concept of plate amortization -see section 2.2.2. Plate amortization is particularly useful in the context of heavily parameterized density approximators such as normalizing flows, but is not tied to it: plate-amortized Mean Field (Blei et al., 2017), ASVI (Ambrogioni et al., 2021a), or implicit (Yin & Zhou, 2018; Titsias & Ruiz, 2019) schemes are also possible to use. Plate amortization can be viewed as the amortization of common density approximators across different sub-structures of a problem. This general concept could have applications in other highly-structured problem classes such as graphs or sequences (Wu et al., 2020; Salehinejad et al., 2018).

### C.4  CONNECTION WITH META-LEARNING

In section 2.2.2 we introduced plate amortization: sharing the parameterization and learning across a model's plates. In this section we discuss the connection between plate amortization and meta-learning(Ravi & Beatson, 2019; Iakovleva et al., 2020; Yao et al., 2019).

Supervised learning can be seen as the mapping from a given context set $\mathcal{C} = \{(x, y)\}$ to a predictive function $f$ (Bishop, 2006) such that $f(x) = y$. Meta-learning –or "learning to learn"– instead recovers this mapping $\mathcal{C} \mapsto f$ in the general case. Once the meta-training is completed, a predictive function $f$ conditioned by an unseen context $\mathcal{C}$ can be obtained in a single forward pass –without any training done on $\mathcal{C}$. As an instance of meta-learning, the Neural Process Family encodes the context $\mathcal{C}$ via a deep set encoder (Garnelo et al., 2018; Dubois et al., 2020; Zaheer et al., 2018). The encoded context, along with the data point $x$ are then used to condition an estimator for the density $q(y|x, \mathcal{C})$.

This framework is similar to the PAVI-E scheme, where a combination of a deep set encoder and a normalizing-flow-based density estimator output the posterior probability of a ground RV $\theta_{i,n}$. This encoder-estimator pair is repeatedly used across a plate. This is analogous to meta-learning to solve the inference problem across the different elements of a plate –such as the different subjects of a population study. In PAVI-E, the encoding $\mathbf{E}_{i,n}$ at a lower hierarchy play a role similar to the context $\mathcal{C}$ in meta-leaning. A few differences however exist between the two frameworks:

- meta-learning is typically concatenated to the 1-plate regime, whereas PAVI is designed for the multi-hierarchy scenario;
- meta-learning typically considers a set of i.i.d. tasks, whereas in PAVI the inference of different ground RVs $\theta_{i,n}$ are conditionally dependent through the hierarchical model $p$;
- meta-learning is trained using the forward KL loss, that is to say in the sample-amortized regime, maximizing the probability $q(y)$ of samples from the underlying generative process. In contrast PAVI –though possible to train using the forward KL loss– is trained via the reverse KL, needing to explicitly evaluate the density of the generative process.

We suspect there would be interesting applications for the PAVI architecture in hierarchical meta-learning scenarios.

## C.5 CONDITIONAL DEPENDENCIES MODELLED IN THE VARIATIONAL FAMILY

The Mean-Field approximation was originally introduced in VI to facilitate computation, allowing dedicated optimization schemes (Blei et al., 2017). Nonetheless, this approximation assumes independence between RVs in the posterior, and ultimately limits the expressivity of the variational family. To remove this approximation, modelling complex conditional dependencies in the variational family is an open research subject (Ambrogioni et al., 2021b; Webb et al., 2018). In the PAVI design, we inherit our statistical dependency structure from the prior distribution $p$, as detailed in eq. (2). This choice of dependencies follows the line of research from Hoffman & Blei (2014) and Ambrogioni et al. (2021a). As pointed out by Ambrogioni et al. (2021b), when modelling only those *forward* dependencies, the modelling of colliders can be an issue.

In the case of a single plate –the 2-level case– Agrawal & Domke (2021) demonstrate that modelling only the *forward* dependencies does not result in reduced expressivity compared to the modelling of the full dependencies. Yet this result does not hold in the n-level case, as Webb et al. (2018) show that faithful inversion features conditional dependencies in the posterior between ground RVs of the same template. We can dub those dependencies as *horizontal* dependencies, across RVs in the same plate. Similar to Structured SVI (Hoffman & Blei, 2014) or Automatic SSVI (Ambrogioni et al., 2021a), the PAVI design therefore results in reduced expressivity when stacking multiple plates in the model $p$. This issue can be partially alleviated with the usage of a *backward* encoding scheme –going in the reverse direction compared to the prior's dependencies– as in the PAVI-E design (see section 2.3.2) or in Cascading Flows (Ambrogioni et al., 2021b).

In practice, though limiting our expressivity, *horizontal* dependencies are difficult to inject back into our architecture. Critically, in the PAVI design, the use of a common density estimator across the ground RVs of the same template (see section 2.2), and the stochastic training over batches of those RVs (see section 2.3.1) prevent the direct modeling *horizontal* dependencies. Put differently, the fact that we consider the inference over different ground RVs as conditionally independent inference problems is central to our design, and adverse to the modeling of *horizontal* dependencies.

Injecting *horizontal* dependencies back into our variational family is therefore a non-trivial research direction, that is not at the core of this paper. This opens up promising research directions: how

could arbitrary conditional dependencies be modelled in the variational posterior in the context of stochastic training?

### C.6 TOWARDS USER-FRIENDLY VARIATIONAL INFERENCE

By re-purposing the concept of amortization at the plate level, our goal is to propose clear computation versus precision trade-offs in VI. Hyper-parameters such as the encoding size –as illustrated in fig. 3 (right)– allow to clearly trade inference quality in exchanged for a reduced memory footprint. On the contrary, in classical VI, changing $\mathcal{Q}$'s parametric form –for instance switching from Gaussian to Student distributions– can have a strong and complex impact both on number of weights and inference quality (Blei et al., 2017). By allowing the usage of normalizing flows in very large cardinality regimes, our contribution aims at disentangling approximation power and computational feasibility. In particular, having access to expressive density approximators for the posterior can help experimenters diversify the proposed HBMs, removing the need of properties such as conjugacy to obtain meaningful inference (Gelman et al., 2004). Combining clear hyper-parameters and scalable yet universal density approximators, we tend towards a user-friendly methodology in the context of large population studies VI.

