# OpenReview forum: "PAVI: Plate-Amortized Variational Inference"
_ICLR.cc/2023/Conference — Submitted to ICLR 2023_

### Official Review · Reviewer_d76j · 2022-10-22

**Confidence:** 2
**Correctness:** 3
**Technical Novelty And Significance:** 2
**Empirical Novelty And Significance:** 2
**Recommendation:** 5

**Clarity, Quality, Novelty And Reproducibility:**

**Clarity**
- as above, I felt the arguments in the main paper can be made more concisely, I felt this was unnecessarily difficult for me to understand. The parts of the SM that I did read I felt were clear and thorough, and I wish the main paper were made more concise and explicit and many of the SM details may be moved to the main paper. (pseudocode, results and baselines for neuroimaging application)

**Novelty**
- I feel the main contributions seem to be the splitting of normalizing flow parameters into $\phi_i$ shared across multuiple RVs and  RV specific parameters $E_{i,n}$, as well as the unbiased mini batching, which I feel are novel, though their significance may not be sufficient for acceptance.

**Reproducibility**
- the SM is extensive and contains many details and pseudocode, I feel the authors have been very thorough and, apart from a few misunderstanding I have, the method and toy experiments appear reproducible.


**Strength And Weaknesses:**

Overall, I struggled with the paper and I have given a low confidence score due to possible misunderstandings.

**Strengths**
- the Supplementary material seems very comprehensive and answered many of the questions I had after reading the main paper.
- the method seems well designed and thought out
- the encoding size experiment provided good confirmation that the degrees of freedom must match the sufficient statistics.

**Weaknesses**

I feel this paper is very hard to recommend for acceptance due to the writing and the somewhat basic numerical evaluation.

I felt the writing and the structure of the paper was very poor, it took me a great deal of effort and multiple rounds of re-reading and re-evaluating my own understanding of the paper before finally appreciating the paper and its contributions (and I have experience in graphical models).
  - I would make section 2.1 into it's own section that really makes the mathematical problem clear. E.g. a plate diagram, the generative model in  equation (1), a real world example with cardinalities and (non-linear) link functions and latent variables. Describe the practicalities of inferring parameters in such a model and why it is impractical scaling with cardinality.
- I don't know if this is standard or not but I was personally thrown off by the terminology "RV template $\theta_i$" and "ground RVs $\theta_{i,n}$" and "grounding a template with cardinalities". Is is possible to reduce notation? E.g. $\mathcal{T}$ is introduced but never used, plates $\mathcal{P}$ is simply indicated by $i$, Card($\mathcal{P}$) is just $N_i^{full}$. Perhaps $N^{full}$ and $N^{redu}$ could be replaced by $N$ and $\tilde{N}$, i.e. use tilde "~" to denote reduced version of various quantities?
- what does $\pi(\theta_{i,n})$ explicitly represent in equations (1) and (2)? I believe they should be precise, known, values if they are conditioning a distribution, but this is seems vaguely defined in the paper. In Ambrogioni 2021b they say "$\theta_j$ is the array of values of the parents of the parents of the $j^{th}$ variable, and $\pi_j(theta_i)$ and $\pi_j()$ is the link function".
- what is the prior for an RV, $p(\theta_{i,n}| \pi(\theta_{i,n}))$? My intuition is that prior is the should be the generative model up to $\theta_{i, n}$ with all of the parent variables marginalized out? Or is it the generative model term with conditioned on given parent values? In the latter case, having such a distribution as the input to a normalizing flow doesn't feel like it would lead to a good posterior approximation, parent values change, prior changes, normalizing flow function is constant but the input changes?
- Section 2.3: please give dimensions of all **X**, and $\theta$ and I assume one row/col of  **X** is a subgroup indicator variable?
- (nitpick) I am not entirely sure I agree with the use of the word "amortization" in the PAVI-F, the fact that $\phi_i$ is shared across all nodes in a layer does not makle inference faster at test time for new nodes? I still have to learn $E_{new}$ from scratch? (aAlthough amortization means "spreading out", in this community, I feel it refers to predicting an approximate posterior in one shot given observed variables.
- the experiments are toy examples and the only non toy example is rather brief with no quantitative results in the main paper
- Lack of Technical Novelty
  - randomly subsampling nodes in the DAG seems like a very simple straightforward idea and I think the unbiasedness is significant and shows that this isn't "just a hack" but a real valid method, however it is rather incremental.
  - claiming to have reduced memory footprint by "using a low dimensional encoding" seems ambitious claim of novelty. While it may be novel, forcing the method to use a low dimensional parameterization of a normalizing flow seems incremental.

- I was surprised that the discussion about "faithful posteriors" (Webb et. el. 2019) was in the SM, to me this felt like very closely related work and provides a method for building approximate posterior Bayesian networks,

**Summary Of The Paper:**

The authors consider approximate inference in Bayesian networks, specifically in the subset of Bayesian networks that can be represented as plate diagrams, which are common for large population studies, the population is composed of subgroups and each group is composed of many individuals. Given global the population parameters, each of the group parameters are i.i.d. and given group parameters, the individuals are i.i.d., a common hierarchical structure.

Assuming a fully specified Bayesian generative model of the data, and only leaf nodes are observed, the authors propose a variational inference method to learn the full approximate posterior distribution to all of the unobserved nodes. The method combines normalising flows with the generative model priors to approximate the posterior. The method is applied to illustrative toy datasets and shows results to be expected and a high dimensional neurological time series dataset (but results are not presented in the main paper).


**Summary Of The Review:**

- clarity of the main paper is very poor, verbose and possibly non-standard terminology for what appears to be standard Bayesian networks.
- the technical contribution, parameterization of normalizing flows and mini batch training, seem novel but incremental.
- the experiments are mostly toy, one non toy is only briefly considered.

UPDATE: I have updated my score from 3 to 5 after reading through the responses.

---

> ### Author Response · Authors · 2022-11-13
> **Detailed answer: d76j (1/2)**
>
> Thank you for the time you invest in this process.
>
> In complement to our general answer, we answer point-by-point to your various concerns below. Points [A], [B], [C] relate to our general answer. Please do not hesitate to answer in this thread if you require further information.
>
> Overall, we apologize for the inconvenience caused by our writing style. We thank you for your proposals to improve the structure of our paper, many of which we propose to implement in a revised manuscript.
>
> ### Rework of section 2.1
>
> Thank you for your recommendation. We argue that the elements you describe (mathematical formulation, generative model, practicalities of large-scale inference) are already present in our manuscript, and that the issue is rather about reorganizing this material, which we propose to do in an updated manuscript.
>
> ### Template versus ground RVs terminology, plates
>
> To our knowledge, this terminology is standard, and is for instance documented in a book from Koller & Friedman [1]. Plates are also referred to in Bishop [2].
>
> Our method revolves around multiple-hierarchy HBMs, parameters sharing and subsampling across “similar” RVs. We argue there is a need for an unequivocal terminology instead of referring to “nodes” or “RVs”. This formalism for instance was deemed interesting in itself by reviewer pbEf. For those reasons we will keep the template terminology.
>
> Regarding operators such as $\text{Card}$ or the tilde notation, we propose to try out those lighter notations in a revised manuscript.
>
> ### $\pi$ notation
>
> As denoted below equation 1, $\pi(\theta_{i,n})$ denote the parents of the RV $\theta_{i,n}$. This is similar to the notation $\pi_j$ in Ambrogioni (2021b), section 3.1, where $\pi_j$ also represents parent RVs. Greek letters such as $\theta$ are to our knowledge oftentimes used to denote RVs [3]. For those reasons we will keep the current notations.
>
> ### Prior for an RV $p(\theta_{i,n} | \pi(\theta_{i,n})$
>
> Your second intuition is correct: $p$ describes here the generative model term conditioned by the values of its parents. This cascading flow is inspired from Ambrogioni (2021b) [4].
>
> As per your intuition, the distribution transformed by the flow depends on the parent values. Taking the example of a Gaussian centered on the parent value, the prior distribution will be a Gaussian of varying mean. This allows us to use the prior as a starting point for the posterior –oftentimes both belong to the same family of distributions. This also allows coupling between the parent’s and child’s samplings, removing independence in the posterior –as per the Mean Field approximation [6].
>
> For a more detailed argument about the interest of that scheme, please refer to the cascading flow paper [4].
>
> ### Dimensions in 2.3
>
> We do not understand this question, please correct us if we misinterpreted it.
>
> Are you referring to a specific HBM? We meant the notations in 2.3 to denote generic observed RVs ($X$) and latent inferred parameter RVs ($\theta$).
>
> If you refer to figure 1, in the full model $\Theta = (\theta_1, \theta_2)$ where  $\theta_1$ would correspond to a “volume” of size $(3, D_1)$, $\theta_2$ would to a volume of size $(1, D_2)$, and X to a volume of size $(6, D_X)$.
>
> ### Definition of amortization
>
> In the PAVI-F case, amortization refers to the fact that the conditional flow $\mathcal{F}_i$ can “one shot” output the posterior for a RV given the corresponding encoding. In your analogy, encodings play the role of observed data. One of our original contributions is to apply amortization to a different granularity.
>
> We benefit from what we learn from the inference of a given RV to infer faster the posterior of another RV, via the shared $\mathcal{F}_i$.
>
> Regarding your question about an unseen RV, nodes are not meant to be “added” during inference, but keeping $\mathcal{F}_i$ and only deriving the encoding $E$ would no doubt quicken the inference for the new node, in a mechanism similar to the stochastic training.
>
> ### Experimental campaign
>
> Please refer to our point [C]. Our supplemental material –that we will better link in a revised manuscript– as well as the specificity of our hierarchical setup –both in terms of examples and baselines– should be factored in. Given those points, we argue that our experimental campaign is carefully crafted and rather comprehensive.

---

> > ### Author Response · Authors · 2022-11-13
> > **Detailed answer: d76j (2/2)**
> >
> > ### Novelty
> >
> > You misinterpret “subsampling nodes” and having a “reduced memory footprint” as being our main claims of novelty. Though both are interesting properties of PAVI, our main claim results from the fruitful combination of amortization and stochastic training, as detailed in point [A]. We perform inference a hundred times faster than baselines, which is a significant improvement worth reporting.
> >
> > ### Comparison to faithful posteriors
> >
> > Faithful inversion [5] relates to the statistical dependencies between RVs in the posterior. The “inverted graph” can be used to connect density estimators in the posterior, similar to replicating the prior’s dependencies in CF [4]. As such, applying faithful inversion would result in the same scaling as CF: roughly linear with the number of RVs.
> >
> > Though it is an excellent resource to analyze the gap of variational families in C.4, faithful posteriors wouldn’t constitute a baseline more adapted to our setup than CF.
> >
> >
> > *[1] D Koller and N Friedman, Probabilistic Graphical Models (2009)*
> >
> > *[2] C M Bishop, Pattern recognition and machine learning (2006)*
> >
> > *[3] As a generic resource, https://en.wikipedia.org/wiki/Notation_in_probability_and_statistics*
> >
> > *[4] Ambrogioni et al., Automatic variational inference with cascading flows (2021b)*
> >
> > *[5] Webb et al., Faithful Inversion of Generative Models for Effective Amortized Inference (2019)*
> >
> > *[6] Blei et al., Variational Inference: A Review for Statisticians (2017)*

---

### Official Review · Reviewer_oFPk · 2022-10-24

**Confidence:** 4
**Correctness:** 3
**Technical Novelty And Significance:** 2
**Empirical Novelty And Significance:** 2
**Recommendation:** 3

**Clarity, Quality, Novelty And Reproducibility:**

Clarity: Heavy notation makes the text hard to follow.

Quality: I have doubts about the theoretical soundness, the experimental campaign is limited.

Novelty: seems incremental compared to ADAVI.

Reproducibility: I have no concerns regarding reproducibility.

**Strength And Weaknesses:**

## Strengths:
* The methodology is generic and intuitive
* Potential for large scale applications
* The idea of embedding structure into the approximate inference scheme is potentially fruitful line of further research
* The authors provide detailed discussion in the appendix, including some limitations regarding the approximation gap.
* The empirical validation in a realistic large scale scenario.

## Weakness:
* Heavy notation makes the text hard to follow;
* Limited novelty (e.g., the work is incremental when compared with ADAVI). It seems that the major difference between PAVI and ADAVI is the stochastic updates of variables;
* I believe the gradient estimates for PAVI are biased;
* Limited experimental campaing, limited to a small set of methods and models. Validation against multiple methods in smaller dataset could be beneficial.

## Questions
1. In the demonstration on Appendix A.2 it indicates that PAVI-F is unbiased by showing in A.12 that the ELBO term is equal the term with all the variables, however Section A.4 implies that there is a gap between vanilla VI, PAVI-F and PAVI-E: Gvanilla VI ≤ GPAVI-F ≤ GPAVI-E . This seems unintuitive and indicative that indeed PAVI-F might be biased, given that the term that differ in those approximation schemes is indeed the ELBO term. Are both of those results consistent?

2. How does the proposed methodology would compared with MCMC/HMC? It could be beneficial given that a new approximation scheme is being proposed to compare it as well sampling methods.

3. When it comes to the expressivity of the variational approximation. How does the predictive performance of the method compare with others variational inference proposals (empirically)? For example another family of expressive VI approximations are found in the works such as UIVI [1], SIVI [2] or VCD [3], which does not use flows and yet are generic and could be a better test bench and give the reader more confidence about the proposed method.

4. In Eq. A.11, when two variables share the same descendants, hence leading to correlated E_{i,n} in the PAVI-E scheme, wouldn’t in this case lead also to dependency between the indicator random variables and log q_{i,n} (given the dependency of log q on the encoding)?


**Summary Of The Paper:**

The work proposes an amortized VI scheme that leverages the hierarchical structure of a Bayesian model, using the latter's plate representation. Plates are used to instantiate subsets of the variables of interest, and share weights in the encoding of sets of variables used on a conditional flow architecture. The result is a plate-amortized proposal distributionI. This idea seeks to enable VI in a large scale setup. The methodology is validated empirically against the state-of-the-art, with one application with real data in neuroimaging context.


**Summary Of The Review:**

Given 1) the limited novelty, 2) short experimental campaign, 3) and possible issues with theory, I recommend rejection.

---

> ### Author Response · Authors · 2022-11-13
> **Detailed answer: oFPk (1/2)**
>
> Thank you for the time you invest in this process.
>
> In complement to our general answer, we answer point-by-point to your various concerns below. Points [A], [B], [C] relate to our general answer. Please do not hesitate to answer in this thread if you require further information.
>
> ### Heavy notation
> We apologize for the inconvenience. Would the notation rework proposed by reviewer d76j (based over tilde notations) be clearer to you?
>
> ### Limited novelty compared to ADAVI
> The differences with the ADAVI [1] architecture are detailed in point [B]. Though measurable, those differences do not constitute our main claim, as underlined in point [A]. PAVI is trained a hundredfold times faster than ADAVI, which is a significant improvement worth reporting.
>
> ### Limited experimental campaign
> Please refer to our point [C]. There is only a limited amount of baselines that can be adapted easily to large scale hierarchical settings. We compare CF and ADAVI in our original manuscripts and we will add UIVI in our revised manuscript, to show the soundness of our method. If we are missing another baseline that can be used in our setting, we will be happy to add it in our experiments.
>
> ### Stochasticity-induced bias versus variational family gap
> Section A.2 refers to the bias introduced training stochastically instead of full-model over the variational family $\mathcal{Q}$. The question is whether we find the same optimal distribution $q^* \in \mathcal{Q}$ by training over subsets of the full model’s graph.
>
> In contrast, in section A.4 the gap $\mathcal{G}$ is an asymptotic bound over the closeness of $q^*$ to the ground truth posterior [2]. $\mathcal{G}$ is a property of the variational family $\mathcal{Q}$, not of the training procedure.
>
> Provided perfect optimization,  $\mathcal{G}$ would be the KL divergence between $q^*$ and the posterior, where $q^*$ is obtained training stochastically over $\mathcal{Q}$. Both sections are consistent.
>
> ### Comparison to MCMC
> We are not convinced that the comparison with MCMC would be beneficial to our paper. A commonplace argument in favor of VI is that sampling-based methods struggle in high-dimensional setups [3]. We reach hundreds and even a million latent parameters in our case. In addition, neither the baselines ADAVI [1], CF [4] nor the baseline UIVI [5] you propose compare themselves to MCMC. Moreover, numerical comparison is not possible with the ELBO as our metric, given that MCMC only defines a distribution implicitly. Lastly, PAVI doesn’t share any methodological similarity with MCMC that would make the comparison insightful.
>
> ### Comparison to Semi Implicit VI [6] / Unbiased Implicit VI [5]
> Thank you for pointing us towards this relevant literature, that we will discuss and cite in an updated manuscript.
>
> We argue that the SIVI [6] / UIVI [5] design does not scale well to large population studies. Indeed, both methods do not factorize the parameter space, which quickly becomes too large for the considered applications. Off-the-shelf SIVI / UIVI are furthermore not amenable to stochastic VI.
>
> Nevertheless, we include UIVI in our comparison for the smaller scale settings, and show that as the dimension of the parameter space grows, so does the parameterization. For large cardinalities, SIVI / UIVI cannot be applied.
>
> Factorizing the parameter space $\theta$, implicit distributions could be used instead of normalizing flows. An amortization scheme should be derived,for instance concatenating the encodings $E$ on top of the UIVI RV $\epsilon$. PAVI would then play the role of a generic scheme to combine hierarchically density estimators. We propose to discuss this point in our appendix.
>
> ### Comparison to Variational Contrastive Divergence [7]
> Due to a missing reference, we interpreted “VCD” as Variational Contrastive Divergence [7]. Please correct us if this assumption is wrong.
>
> If our assumption is correct, thank you for pointing us towards this relevant literature, that we will discuss and cite in an updated manuscript.
>
> It is our understanding that VCD plays the role of a loss to optimize a given variational family. As such, VCD could be applied to any of the baselines we compare, but VCD in itself has no particular reason to be compared to PAVI.

---

> > ### Author Response · Authors · 2022-11-13
> > **Detailed answer: oFPk (2/2)**
> >
> > ### Eq. A.11: 2 RVs with colliders in the PAVI-E scheme
> > We assume this question is related to your argument about biased gradient estimates. Please correct us if this assumption is wrong.
> >
> > You argue the existence of a correlation between the fact that those RVs are selected at a stochastic training step, and the value of their encoding.
> > If 2 RVs $(\theta_1, \theta_2)$ share a child $\theta_3$, following A.1.1 they correspond to the same plate indices. $(\theta_1, \theta_2)$ will be both selected or both not selected at a given stochastic step. Then:
> > - If the child $\theta_3$ is unique, it must belong to the same plates as its parents: the encoding is a constant.
> > - If there are multiple iid children as part of a plate, the value of the encoding will vary depending on the selected minibatch of those children. This minibatch will depend on the index of the plate $\theta_3$ belongs to but not $(\theta_1, \theta_2)$. So the indicator of the selection of $(\theta_1, \theta_2)$ and the derivation of the encoding for $q$ rely on independent plate indices.
> >
> > In all cases, A.11 is correct. Note that this does not imply that the PAVI-E scheme can be proven unbiased, as detailed in A.2.3.
> >
> > *[1] Rouillard & Wassermann, ADAVI: Automatic Dual Amortized Variational Inference Applied To Pyramidal Bayesian Models (2022)*
> >
> > *[2] Cremer et al., Inference Suboptimality in Variational Autoencoders (2018)*
> >
> > *[3] Blei et al., Variational Inference: A Review for Statisticians (2017)*
> >
> > *[4] Ambrogioni et al., Automatic variational inference with cascading flows (2021b)*
> >
> > *[5] Titsias et Ruiz, Unbiased Implicit Variational Inference (2019)*
> >
> > *[6] Yin et ZHou, Semi-Implicit Variational Inference (2018)*
> >
> > *[7] Ruiz et Titsias, A Contrastive Divergence for Combining Variational Inference and MCMC (2019)*

---

### Official Review · Reviewer_pbEf · 2022-10-26

**Confidence:** 4
**Correctness:** 3
**Technical Novelty And Significance:** 2
**Empirical Novelty And Significance:** 2
**Recommendation:** 6

**Clarity, Quality, Novelty And Reproducibility:**

Clarity and quality: I think the paper is well-written in general, although there are some complex notations to digest which would be hard to fully grasp at a first glance.
Novelty: as mentioned above, it is worth discussing the relevant works in meta-learning literature, and clarifying the novelty of the proposed method compared to them.
Reproducibility: I think the paper provides enough details to reproduce the experiments.

**Strength And Weaknesses:**

Strengths
- The paper is well-written and easy to follow.
- The description of amortization using the language of plate diagram is interesting.
- The neuroimaging experiment seems interesting.

Weaknesses
- In my opinion, the notion of plate amortization is not entirely new, so I think the paper can be strengthened by discussing relevant existing works.
- The experiments include a synthetic Gaussian model and neuroimaging experiments. The synthetic experiment looks promising, but it is of a small scale that does not particularly highlight the benefit of the proposed method. For the neuroimaging experiment, although it seems promising that only the proposed method was manageable to train, it is not clear whether the prediction given by the model is reasonable (at least for the non-experts like me).

Detailed comments
- The idea of plate amortization is indeed interesting. I guess the proposed framework virtually extends to hierarchical Bayesian models with an arbitrary level of hierarchies, but the main application would be the amortization across populations (datasets). Regarding amortization over datasets, there have been similar ideas (although motivated differently), especially in the context of meta-learning. Neural processes families, for instance, introduce an amortized inference network that can readily generalize to a new task (corresponding to the population) which is parameterized by the dataset encoding networks (deep sets or transformers); this would roughly correspond to PAVI-F, except that PAVI-F uses normalizing flows for the parameterization. Amortized Bayesian meta-learning (Ravi and Beatson, 2019) is also worth discussing in a similar sense. Depending on the application, neural processes can be considered as a baseline to be compared to the proposed method, since it can also scale to large population study due to its use of the amortized inference network.
- The experiment section could be enhanced. I understand that the neuroimaging experiment can be compelling, but in the current form, I'm not entirely sure how well the proposed method is doing; the only result I could see from the paper is Figure 5, which I fail to interpret. It would be great to see the numbers, for instance, metrics could tell us how accurate PAVI is, or at least how well the predictions given by PAVI match the domain knowledge given by experts. Also, if the baselines are not trainable due to their limited scalability, one could consider comparing them on a smaller subsets of manageable sizes to see if PAVI would do better than baselines for real-world tasks.

**Summary Of The Paper:**

This paper proposes Plate-Amortized Variational Inference (PAVI), an inference framework for hierarchical Bayesian models with amortization over the population (dataset) level implemented with normalizing flows with shared parameters. The main idea is to posit an expressive variational family parameterized with normalizing flows, and yet reduce the number of parameters to be trained by sharing part of the variational parameters across populations. Another important concept is stochastic training, where for each step, instead of training all the variables present in the graphical model, subsample a small number of variables and compute stochastic gradient with them. Thanks to amortization, PAVI could scale to large-scale population studies involving high-dimensional parameters.

**Summary Of The Review:**

Overall, the proposed method is an interesting contribution, but the paper could be strengthened by adding more discussions and hopefully more experimental results/interpretations.

---

> ### Author Response · Authors · 2022-11-13
> **Detailed answer: pbEf**
>
> Thank you for the time you invest in this process.
>
> In complement to our general answer, we answer point-by-point to your various concerns below. Points [A], [B], [C] relate to our general answer. Please do not hesitate to answer in this thread if you require further information.
>
> ### Novelty
>
> Our main claim does not pertain to our improvements over amortization nor stochastic training alone, but to our original and fruitful combination of the 2, as detailed in point [A]. We perform inference a hundred times faster than baselines, which is a significant improvement worth reporting
>
> ### Synthetic experiments do not highlight PAVI’s benefit
>
> Our experiment 4.3 shows that PAVI matches the inference quality of state-of-the-art baselines, with constant parameterization and constant training time as the cardinality of the problem augments. Obtaining valid posteriors in a hundred times shorter inference time constitutes a clear improvement in our opinion. If the reviewer disagrees, we kindly ask him to point us toward how to better highlight the benefits of PAVI.
>
> In addition, PAVI’s superior inference quality is also illustrated in our synthetic experiments B.3.1, B.3.2 and B.3.3.
>
> ### Validation of Neuroimaging results
>
> Validating a parcellation is a complex and open research topic [1]. Our results match current knowledge from domain experts: we illustrate the functional partition of Broca’s area in lexical and semantic-associated regions [2, 3]. This constitutes a preliminary but already sensical result. Further  validation would require a more involved description of the neuroscientific background, which would go beyond the scope of this  methodological conference paper.
>
> PAVI’s performance is also evaluated numerically on a synthetic version of manageable size of our Neuroimaging model in B.3.3.
>
> ### Amortization across datasets as main application
>
> We respectfully disagree with that statement. As detailed in point [A], the main goal of PAVI is to amortize the inference within a given dataset, and not across datasets. Though sharing many similarities, PAVI cannot be subsumed in the meta-learning framework.
>
> Amortizing across datasets would rather correspond to sample amortization –as defined in 2.3– which is permitted by our architecture, but not necessary. As an example, our Neuroimaging experiment features amortization within a connectome, but not across connectomes: we seek to regularize our results across subjects.
>
> ### Comparison to the Neural Process Family [4, 5]
>
> Thank you for pointing us towards this very relevant literature, that we will discuss and cite in an updated manuscript.
>
> PAVI-E and the NPF [4, 5] share architectural similarities. PAVI’s observed data encoder roughly corresponds to NP’s context set encoders, and PAVI’s normalizing flows could be used in lieu of NP’s parametric distributions –such as Gaussians. We note however that in the context of meta-learning NPs are limited to the 1-plate regime.
>
> Extending NPs to the hierarchical regime is not trivial. As a thought exercise, extending the notion of context to the hierarchical regime, and using NFs as parametric distributions, NPs would correspond to the ADAVI baseline [8], trained using the forward KL loss –using samples from the HBM but not its density function.
>
> In the ADAVI paper, authors extensively show that the training using the forward KL yields significantly worse results in high-dimensional cases. We perform better than ADAVI trained with the reverse KL loss, so it is safe to assume we would perform favorably compared to NPs –even extended to the hierarchical setup.
>
> ### Comparison to Amortized Bayesian Meta-Learning [6]
>
> Thank you for pointing us towards this relevant literature, that we will discuss and cite in an updated manuscript.
>
> As with the NPF, ABML [6] has been designed with the low-hierarchy case in mind. Extension to higher hierarchies would be complex, relying on higher-order derivatives of the loss (beyond the Hessian), and removing the dirac approximation for the global parameters.
>
> ABML extended to the hierarchical setup would equate to an N-level optimization, which is an open research topic [7]. For those reasons, we argue that extending ABML to our setup is beyond what is reasonable to ask from a conference paper.

---

> > ### Author Response · Authors · 2022-11-13
> > **References**
> >
> > *[1] Eickhoff, Yeo and Genon, Imaging-based parcellations of the human brain (2018)*
> >
> > *[2] Heim et al., Effective connectivity of the left BA 44, BA 45, and inferior temporal gyrus
> > during lexical and phonological decisions identified with DCM (2009)*
> >
> > *[3] Zhang et al., Connecting concepts in the brain by mapping cortical representations of semantic relations (2020)*
> >
> > *[4] Dubois et al., Neural Process Family (2020)*
> >
> > *[5] Jha et al., The Neural Process Family: Survey, Applications and Perspectives (2022)*
> >
> > *[6] Ravi et Beatson, Amortized Bayesian Meta-Learning (2019)*
> >
> > *[7] Rommel et al., Deep invariant networks with differentiable augmentation layers (2022)*
> >
> > *[8] Rouillard & Wassermann, ADAVI: Automatic Dual Amortized Variational Inference Applied To Pyramidal Bayesian Models (2022)*

---

> > > ### Comment · Reviewer_pbEf · 2022-11-29
> > > **Response to response**
> > >
> > > Thank you for your detailed response and effort to enhance the readability.
> > > For me, this paper is still borderline, but slightly towards acceptance. I see the proposed framework as a somewhat incremental generalization of existing amortization (within datasets + across datasets) for an arbitrary hierarchical Bayesian model; the concept itself may be new, but not astonishingly surprising. The stochastic training scheme may also be considered novel, but it is also a natural way to go one might think of. But I agree with the authors that the main novelty of the proposed framework comes from their combination, so my overall judgment is leaning toward acceptance.
> > >
> > > The reason why I don't argue for acceptance is mainly the weak experiments; I guess there are some relevant baselines (e.g., NPF or ABML) to be compared, and more experiments (large scale as neuroimaging, but still more interpretable) would convince the strength of the proposed framework. Although I raise my score to weak accept, I will follow the other reviewers' general consensus.

---

> > > > ### Author Response · Authors · 2022-11-30
> > > > **Thank you for your answer**
> > > >
> > > > Dear reviewer,
> > > >
> > > > **Thank you for your engagement in this review process and your positive comments.**
> > > >
> > > > As you point out, the main novelty of our paper doesn’t consist in our generalization of amortization nor stochastic training alone, but in their combination. We underline that this combination doesn’t only combine the advantages of both, which would result in **reduced parameterization and memory footprint *at the cost of slower inference*** (as in experiment 3.1). On the contrary, we show that our design results in **faster inference**: we argue this is an original result, which could be of broad interest in large-scale applications.
> > > >
> > > > Regarding your comment on baselines, we restate our argument (visible in our previous answer). Generalizing ABML or NPF to the hierarchical case is not trivial, and underlines that **we unlock inference for regimes that are not tackled by the state of the art**. Nevertheless, the NPF is close in spirit to the baseline ADAVI, and the latter can be considered as an upper bound for the performance of the former, as demonstrated by Rouillard et Wassermann in the ADAVI paper.
> > > >
> > > > **In conclusion, we unlock inference in a very efficient fashion in the previously unattainable large-scale regime, which we argue is novel.**
> > > >
> > > > We stand at your disposal should you have additional comments,
> > > >
> > > > Best regards,
> > > >
> > > > The authors

---

### Official Review · Reviewer_1MMm · 2022-10-28

**Confidence:** 3
**Correctness:** 3
**Technical Novelty And Significance:** 3
**Empirical Novelty And Significance:** 3
**Recommendation:** 5

**Clarity, Quality, Novelty And Reproducibility:**

* **Clarity:** The paper is quite dense to read. For instance, sections 3.1 and 3.2 seem to be repeating what is in Sec. 2.3 and 2.4 at times perhaps a bit too much. Something more interesting would be to expand on the unbiasedness of the approximation, which has been left for the appendix (A.2).

* **Novelty:** I've found some of the methodology somewhat similar to that of ADAVI (Rouillard & Wassermann, 2022). The difference lies on the use of conditional independence structures between parameters to further factorise the posterior.

* **Reproducibility:** Code is provided and the dataset is open access.


**Strength And Weaknesses:**

### Strengths
* The proposed method achieves significant (orders of magnitude) performance speed-ups when compared to baselines in experiments.
* Although the idea of using factorised variational posteriors for hierarchical models is not new, this paper implements it in a more computationally efficient way, which is also amenable for per high-performance computing with hardware accelerators.
* Diagrams help understand the concepts in the approach.

### Weaknesses

* Related work: There are other flexible variational inference approaches for hierarchical models which also factorise the variational posterior. For example, Tran et al. (2017) introduced hierarchical implicit models for this task, and other relatively simple approaches for random effects models (as in the example experiment) are also available (e.q., Dao et al., 2022).

* Experiments: There was only one benchmarking model (Eq. 6) and variations thereof for the comparisons. The other experiment (Fig. 5) did not provide comparisons against baselines due to its complexity. Another representative set of experiments would help assessing the improvements over baselines in corner cases where other methods can be applied but PAVI should in principle perform better.

#### References
Tran, D., Ranganath, R., & Blei, D. M. (2017). Hierarchical Implicit Models and Likelihood-Free Variational Inference. 31st Conference on Neural Information Processing Systems (NIPS).

Dao, V. H., Gunawan, D., Tran, M., Kohn, R., Hawkins, G. E., & Brown, S. D. (2022). Efficient Selection Between Hierarchical Cognitive Models: Cross-Validation With Variational Bayes. Psychological Methods, (April).

**Summary Of The Paper:**

This paper presents an approach for variational inference on large-scale hierarchical models which takes advantage of conditional independence in plate structures to factorise its variational approximations and better scale with problem dimensionality. The proposed framework is based on normalising flows taking shared parameters and data-encoding vectors. The resulting algorithm is equipped a stochastic optimisation scheme for training and is shown to converge faster than other algorithmic baselines in experiments.

**Summary Of The Review:**

The paper's contributions are somewhat novel, but it lacks in experimental comparisons against the baselines in other challenging scenarios.

---

> ### Author Response · Authors · 2022-11-13
> **Detailed answer: 1MMm**
>
> Thank you for the time you invest in this process.
>
> In complement to our general answer, we answer point-by-point to your various concerns below. Points [A], [B], [C] relate to our general answer. Please do not hesitate to answer in this thread if you require further information.
>
> ### Comparison to Hierarchical Implicit Models [1]
>
> Thank you for pointing us towards this relevant literature, that we will discuss and cite in an updated manuscript.
>
> HIM [1] is fitted for the 1-plate case. As is, none of our examples contain only 1 plate. More importantly, HIMs are constructed for the likelihood-free setup, defining the likelihood $p$ and the variational $q$ only through samples, and relying on the density ratio trick [2] to estimate gradients of the KL objective.
>
> A parallel can be made between HIMs and Simulation-Based Inference [2] and in particular the SNRE architecture [3]. Rouillard & Wassermann compared explicitly to likelihood-free architectures in the ADAVI paper [4]. They showed that likelihood-free architectures yielded worse performance in high-dimensional cases such as ours. As such, even extended to the hierarchical case, HIM isn’t likely to perform competitively in our experiments.
>
> ### Comparison to manual VI [5]
>
> Thank you for pointing us towards this relevant literature, that we will discuss and cite in an updated manuscript.
>
> The work from Dao et al. [5] constitutes an excellent example of carefully-crafted variational families, with gradients derived manually. This methodology is likely to yield good performance in problems it can be applied to, but this requires technical mastery and additional derivations should a new HBM be studied.
>
> In contrast, PAVI is in the line of Automatic VI [6, 7, 8]. There is a need for generic variational methods that can be applied to any HBM $p$ and can generate an efficient variational family by automatically analyzing $p$. As such, our methods are instances of automatic VI, and the models proposed by Dao et al. [5] would not fit in that category.
>
> ### Experimental campaign
>
> Please refer to our point [C]. Our supplemental material –that we will better link in a revised manuscript– as well as the specificity of our hierarchical setup –both in terms of examples and baselines– should be factored in.
>
> Regarding corner cases where PAVI shows superior performance, PAVI yields superior inference quality in B.3.1, B.3.2 and largely superior performance in B.3.3 –a numerical comparison over the model used in our figure 5. What’s more, in our experiment 4.3, PAVI shows similar performance to baselines, but with orders of magnitude faster convergence. This constitutes our main claim, which is not necessarily to improve inference quality but to scale up inference.
>
> ### Merging parts of section 2 and 3
>
> We assume you propose to merge the 2 separate mentions of encoding schemes (2.4 and 3.2), which we propose to do in an updated manuscript. However it is unclear to us which information you think is repeated in sections 2.3 and 3.1. With the current text, we separate the design of the variational family and of the training scheme. This supports our narrative that combining amortization and stochastic training non-trivially yields dramatically faster inference.
>
> Could you maybe specify which information you think is redundant?
>
> ### Limited novelty compared to ADAVI
>
> The differences with the ADAVI [4] architecture are detailed in point [B]. Though measurable, those differences do not constitute our main claim, as underlined in point [A]. PAVI is trained a hundredfold times faster than ADAVI, which is a significant improvement worth reporting.
>
> *[1] Tran et al., Hierarchical Implicit Models and Likelihood-Free Variational Inference (2017)*
>
> *[2] Cranmer et al., The frontier of simulation-based inference (2020)*
>
> *[3] Hermans et al., Likelihood-free MCMC with Amortized Approximate Likelihood Ratios (2020)*
>
> *[4] Rouillard & Wassermann, ADAVI: Automatic Dual Amortized Variational Inference Applied To Pyramidal Bayesian Models (2022)*
>
> *[5] Dao et al., Efficient Selection Between Hierarchical Cognitive Models: Cross-validation With Variational Bayes (2022)*
>
> *[6] Kucukelbir et al., Automatic Differentiation Variational Inference (2016)*
>
> *[7] Ambrogioni et al., Automatic structured variational inference (2021a)*
>
> *[8] Ambrogioni et al., Automatic variational inference with cascading flows (2021b)*

---

### Author Response · Authors · 2022-11-13
**General answer**

**We thank the reviewers for their valuable time and advices.** Our method PAVI was deemed intuitive and well designed (oFPk, d76j), and all reviewers appreciated the importance of providing inference methods amenable to large scale. PAVI demonstrates potential for large cardinality applications (oFPk) due to its theoretical properties, but also its efficient implementation (1MMm). This potential is underlined via an original and realistic Neuroimaging experiment (oFPk, pbEf). **Overall, the reviewers share a positive view on the method’s ability to tackle challenging experimental setups.**

We identified the main points of concern across reviewers, for which we provide general answers below. More detailed comments on points [A], [B], [C] are posted as threads of this message to keep the overall discussion concise.

## [A] Novelty

Reviewers have identified PAVI’s amortization and stochastic training schemes separately as our main contributions. Though measurable, those improvements may only seem marginally novel.

PAVI’s novelty does not reside in those improvements taken separately, but in their combination. Their combination is non-trivial, as they can be combined in a number of ways. We derive a scheme that results in orders of magnitude faster inference  –as seen in our scaling experiment 4.3.

This **quicker convergence is a strong improvement over the state-of-the-art, and opens an avenue to apply HBM and VI to large scale problems**, which currently cannot be tackled by amortized nor stochastic architectures alone.

We propose to better state our main claim in our introduction as well as in our methods.


## [B] Comparison to ADAVI

Similar to ADAVI [1], PAVI tackles population studies. While both architectures share similarities, PAVI presents **multiple significant improvements** over the ADAVI architecture:
- tackling generic generative models
- removing the Mean Field approximation in favor of the cascading flow scheme
- introducing several variant encoding schemes, making it amenable to stochastic training

Yet those improvements do not constitute the main claim of our paper, as per our point [A].

We propose to better underline the difference between both architectures in our baselines paragraph section 4.3.

## [C] Experimental campaign

Our experiments in section 4 relate to the main claim of PAVI: orders-of-magnitude faster inference (4.1), which becomes more interesting as the cardinality of the problem augments (4.3). Those core results are largely complemented in our supplemental material, with experiments evaluating **theoretical** properties of PAVI (B.2), and a range of **comparative** experiments with baselines (B.3).

What’s more, PAVI tackles an interesting class of problems –population studies– that yet **does not feature canonical evaluation examples.** We tackle this issue by proposing a variety of HBMs validating complex cases of inference.

It appears that several reviewers missed our experiments in the supplementary materials. We will better link the extra experiments in our main text, in particular our numerical comparisons B.3.


## [D] Writing

Different opinions were expressed regarding the writing of the paper, deemed both well-written and easy to follow by reviewer pbEf yet very poorly written by reviewer d76j. We propose concrete axes for improvement in our answers to each reviewer.

## [E] Additional baselines

PAVI focuses on solving inference in the context of population studies. As such, PAVI has been designed with multiple hierarchies in mind. We appreciate the **many baselines** proposed by the reviewers, that we will discuss and cite in our revised manuscript. We discuss specific methods in our answer to each reviewer.

However, though most **proposed baselines are adapted to low-hierarchy scenarios**, they are not usable off-the-shelf for higher hierarchies. In most cases, **the extension to the hierarchical case is not trivial**, and for instance constitutes the novelty of the baseline ADAVI [1]. We provide concrete arguments for each baseline as to why their extension to the hierarchical case –both theoretically and in terms of code– goes **beyond the scope of a single conference paper**.
Only UIVI [2] can be used as is in our examples. We have implemented this baseline and will add it to our numerical comparisons in our revised manuscript.

*[1] Rouillard & Wassermann, ADAVI: Automatic Dual Amortized Variational Inference Applied To Pyramidal Bayesian Models (2022)*

*[2] Titsias et Ruiz, Unbiased Implicit Variational Inference (2019)*

---

> ### Author Response · Authors · 2022-11-13
> **[A] Novelty- detailed argument**
>
> We bring novel contributions in terms of amortization and stochastic training, yet our main claims pertain to our original combination of those two points.
>
> Both amortization and stochastic optimization are commonplace in the literature [1, 2]. A case can be made on our original **conceptualization of both amortization and stochastic training via the plate formalism**. This in itself is novel and was for instance deemed interesting by reviewer pbEf. The plate formalism brings various advantages:
> - It makes our method generic and automatically applicable to hierarchical Bayesian models with an arbitrary level of hierarchies (oFPk, pbEf);
> - It results in the usage of amortization at a different granularity, with multiple encapsulated schemes, which is to our knowledge novel. Contrary to most instances of amortization, **we do not amortize across different inference problems, but within a given problem**;
> - It helps with the design of a computationally efficient design, via the fast slicing of encoding arrays fed to density estimators on GPU-accelerated hardware (1MMm).
>
> Those improvements have been deemed only marginally novel by reviewers oFPk & d76j, who questioned PAVI’s novelty. Yet those **arguments about novelty misinterpret our main claim, which does not pertain to those improvements taken individually, but to their fruitful combination**.
>
> Our major claim lies within the faster inference yielded by our design. We **harnessed two usually detrimental properties to gain a strong computational advantage**:
> - In itself, amortization hurts the inference quality for a given data point. As detailed in our appendix A.4, amortization can only result in a worse variational posterior compared to optimizing directly over the observed data;
> - Similarly, stochastic training results in slower variational inference: only a few variational weights are optimized at a given step, and gradient estimates can suffer from larger variance in hierarchical cases.
>
> Yet, as demonstrated in our experiments 4.1, 4.2, B.2 and B.3, not only does our stochastic method match the inference quality of full-model baselines, but its convergence is orders of magnitude quicker.
>
> Without speeding up inference, solving extremely large problems such as our Neuroimaging experiment is computationally intractable. Obtaining inference results faster allows to speed up the experimental cycle from a week to an hour. **PAVI unlocks fast inference for extremely large models, which is a strong and novel claim**.
>
> We propose to better state this claim in our introduction as well as in our methods.
>
> *[1] Hoffman et al., Stochastic variational inference (2013)*
>
> *[2] Zhang et al., Advances in Variational Inference (2019)*

---

> ### Author Response · Authors · 2022-11-13
> **[B] Comparison to ADAVI - detailed argument**
>
> ADAVI [1] is a strong baseline architecture for all our experiments. As underlined in our introduction, ADAVI tackles the **over-parameterization problem** in the context of pyramidal models, but cannot deal with very large inference problems due to **computational limits**.
>
> Thanks to its training scheme, PAVI overcomes those computational limits. But more importantly, combining amortization and stochastic training, PAVI gains a **novel property: fast inference**, with a constant training time as the cardinality of the problem augments. This is a qualitative property the ADAVI architecture does not possess.
>
> In terms of detailed differences:
> - ADAVI is limited to the **pyramidal class** of generative HBMs, while ADAVI can tackle any plate-enriched HBM. This includes an arbitrary number of intertwined plates, and observed RVs at higher hierarchies in the graph. This difference for instance allows PAVI to tackle the Neuroimaging model (see B.3.3), which ADAVI cannot do;
> - ADAVI is hampered by the **mean-field approximation**, which is identified by the authors as one of the main limitations of the architecture. In contrast, PAVI benefits from the cascading flow scheme, modeling dependencies in the posterior, and using prior distributions as warm-ups for posterior density estimation. The cascading scheme also enables the presence of observed RVs not located at the leaves of the dependency graph, which is something ADAVI cannot deal with. This difference also partially explains superior performance in our experiment B.3.2 featuring a complex posterior;
> - ADAVI is limited to **full-model, sample-amortized inference**. This results in an amortization gap which can be particularly harmful in large-dimensionality contexts, with a single data point observed. This is further detailed in our appendix A.4. In contrast, PAVI presents a variety of schemes, and details how those schemes interact with stochastic training, a non-trivial extension notably for the PAVI-E scheme
>
> To conclude, though sharing architectural similarities, **PAVI is massively faster to train than ADAVI**, as shown in experiment 4.3. This is not a trivial result: training stochastically a given variational architecture typically results in slower convergence, as detailed in point [A]. **PAVI therefores improves on many architectural limitations of ADAVI, but more importantly is drastically faster to train.**
>
> We propose to better underline the difference between both architectures in our baselines paragraph section 4.3.
>
> *[1] Rouillard & Wassermann, ADAVI: Automatic Dual Amortized Variational Inference Applied To Pyramidal Bayesian Models (2022)*

---

> ### Author Response · Authors · 2022-11-13
> **[C] Experimental campaign - detailed argument**
>
> Both our **supplemental material** –that we will better link in our main text– and the **specificity of our hierarchical setup** should be factored in when assessing our experimental campaign.
>
> We argue to have conducted a **large variety of carefully crafted experiments**, ranging from theoretical validation to numerical comparison to baselines. We propose to better **link those experiments** –in part located in our supplemental material– inside our main text:
> - the effects of the **stochastic training’s minibatch size** on the optimization, showing the interesting PAVI-E property to train with a speed independent on the minibatch size (B.2)
> - a classical **mixture model** (B.3.1), showing superior expressivity for PAVI compared to state-of-the-art baselines
> - a hierarchical model  involving **higher-order moments** (B.3.2), showing against baselines the resiliency of PAVI when needing to aggregate complex summary statistics
> - a comparison of PAVI against baselines on our **Neuroimaging HBM**, validating superior performance for PAVI in a dimensionality adapted to the limited scalability of those baselines
>
> This number of experiments is well on par with the current literature. When considering additional experiments, reviewers should take into account the **specific hierarchical regime tackled by PAVI**. Most canonical experiments in the connected literature are limited to a **single-plate** regime, or to **generative modeling** problems that are better suited to implicit distribution architectures. Those regimes have been extensively studied, and aren’t the focus of this work.
>
> In contrast, PAVI tackles **multi-hierarchy** generative models. Those are interesting problems, as population studies are pervasive in experimental science. We regret the **absence of canonical evaluation models** in this scenario, and actually propose several HBMs to play this role. On the contrary, we argue that placating our method over ill-suited if canonical problems would be of limited scientific interest.

---

### Author Response · Authors · 2022-11-18
**Rebuttal revision**

We renew our **thanks** towards reviewers for their valuable time and advice.

Many helpful recommendations were made by all reviewers to improve our manuscript. We **carefully followed those recommendations** and:
- reworked our introduction to discuss the proposed **baselines**
- streamlined our **methods**, with a section 2.1 dedicated to a clearer problem statement and merging mentions of PAVI encoding schemes into section 2.4
- simplified our **notations**, removing heavy `full` and `redu` superscripts
- clarified our **main claim** in our introduction as well as in our methods. PAVI's main claim lies in our original combination of amortization and stochastic training, which results in significantly faster inference
- performed **additional experiments** with the UIVI baseline, visible in 3.3 and B.3
- better **linked our supplemental experiments** in our main text
- clarified our **novelty** compared to the ADAVI baseline
- added clarifying details in our supplemental sections A.2 (stochastic training) and A.4 (inference gap)
- added a supplemental discussion relative to meta-learning: C.4

To ease the review of those improvements, **we have colored in light blue the main differences** in the text.

In conjunction with our general and detailed answers, we argue to have **consistently answered to all of the reviewer’s comments**. Every concern has been addressed either by a modification of our manuscript or via a detailed comment.

We stand at the reviewer’s disposal should they have additional remarks.

---

### Author Response · Authors · 2022-11-28
**We would happily discuss our new material**

Dear reviewers,

We **thank you** again for your time, and your helpful advice which led to our **revised manuscript**. We respectfully argue that we submitted new material that could lead you to update the scoring of our paper:
- Our general and detailed answers **respond point-by-point to your individual concerns**. In particular, we have provided you with detailed arguments related to:
  - [A] our unequivocal novelty in terms of large-scale computational efficiency;
  - [B] our numerous improvements over the baseline ADAVI;
  - [C] our well-developed campaign featuring 8 experiments.

We restate that our main claim, **combining amortization and stochastic training in an original scheme, leads to orders of magnitude faster inference**. This is not a trivial result: stochastic training usually results in slower inference. In contrast, **PAVI leverages amortization to flip stochasticity to an advantage** in terms of inference speed. This argument is well supported by a variety of experiments on a variety of generative models (in sections 3.3 and B.3);
- Our **revised manuscript addresses many of your original concerns over our paper**. We reworked and simplified our methods, clarified our main claim, performed additional experiments –including another baseline– and we integrated and discussed the relevant works you  pointed out in your first review. Thanks to your help, we argue that our new manuscript is more streamlined and readable.

We would gladly discuss this new material with you, and answer any remaining concerns you would have over our method.

Best regards,
The authors

---

### Decision · Program_Chairs · 2023-01-20

**Decision:**

Reject

**Justification For Why Not Higher Score:**

In total, the experimental breadth, the writing, and more detailed discussion of bias in the main paper makes me think the paper needs more work.

**Justification For Why Not Lower Score:**

N/A

**Metareview: Summary, Strengths And Weaknesses:**

The paper aims to speed up and reduce the memory footprint variational inference in hierarchical Bayesian models. The technique proposed is to use plate amortization to share a single distribution across multiple random variables. The shared distribution takes an encoding vector to allow for the distribution to not be the same for different random variables. Multiple encoding schemes are considered. The results look good to a relatively simple baseline of raw stochastic variational inference and several flow based approaches.

The reviewers were negative on the paper initially. This changed to a mixed leaning negative after the author response. With authors moving from reject to borderline reject. One concern was clarity in the notation which was raised by several reviewers. Another concern that was partially addressed in the revision was on the breadth of the experiments, though I still find them still a bit lacking. There was a discussion about bias that is important and brought up by the reviewers, but is buried in appendix a.2.3. In total, the experimental breadth, the writing, and more detailed discussion of bias in the main paper makes me think the paper needs more work.

Beyond those concerns, there are still missing related works such as

@inproceedings{hoffman2015structured,
  title={Structured stochastic variational inference},
  author={Hoffman, Matthew D and Blei, David M},
  booktitle={Artificial Intelligence and Statistics},
  pages={361--369},
  year={2015}
}

and some minor bits on the definitional correctness on what stochastic variational inference is. The cited Hoffman paper (figure 4 or equation 25) propose maximizing over local variational parameters and never storing them, while the stochastic variational inference described is one that explicitly stored.